# Minimal genetically encoded tags for fluorescent protein labeling in living neurons

Aleksandra Arsić [1,2], Cathleen Hagemann [2], Nevena Stajković [1,2], Timm Schubert [1,3] &
Ivana Nikić-Spiegel [1✉]

Modern light microscopy, including super-resolution techniques, has brought about a demand for small labeling tags that bring the fluorophore closer to the target. This challenge can be addressed by labeling unnatural amino acids (UAAs) with bioorthogonal click chemistry. The minimal size of the UAA and the possibility to couple the fluorophores directly to the protein of interest with single-residue precision in living cells make click labeling unique. Here, we establish click labeling in living primary neurons and use it for fixed-cell, live-cell, dual-color pulse–chase, and super-resolution microscopy of neurofilament light chain (NFL). We also show that click labeling can be combined with CRISPR/Cas9 genome engineering for tagging endogenous NFL. Due to its versatile nature and compatibility with advanced multicolor microscopy techniques, we anticipate that click labeling will contribute to novel discoveries in the neurobiology field.

[1] Werner Reichardt Centre for Integrative Neuroscience, University of Tübingen, Otfried-Müller-Straße 25, 72076 Tübingen, Germany. [2] Graduate Training Centre of Neuroscience, International Max Planck Research School, University of Tübingen, Otfried-Müller-Straße 27, 72076 Tübingen, Germany. [3] Institute for Ophthalmic Research, University of Tübingen, Elfriede-Aulhorn-Straße 7, 72076 Tübingen, Germany. ✉email: ivana.nikic@cin.uni-tuebingen.de

Fluorescence microscopy encompasses experimental techniques that are central to the study of cellular and molecular (neuro)biology. These techniques allow the investigation of protein localization and interactions, as well as the visualization of protein dynamics in real time. The ability to investigate these processes in living cells and with high resolution is especially relevant for polarized cells, such as neurons. Aggregation of proteins and perturbation of protein processing, turnover, and axonal transport mechanisms, have been implicated in a number of neurological diseases. Advanced light microscopy techniques, including live-cell and super-resolution imaging are indispensable for elucidating the underlying molecular mechanisms, but to fully utilize their potential, advances in imaging technologies need to be matched with advancement in the field of protein labeling[1–5].

Breaking the resolution limit in modern super-resolution microscopy techniques has brought about a demand for small labeling tags that bring the fluorophore closer to the target. Consequently, there is a trend towards using smaller nanobodies, affimers, and aptamers instead of the conventional primary–secondary antibody complexes[6–8]. Similarly, in live-cell labeling, relatively large fluorescent proteins (FPs) are being substituted with small fluorescent dyes[9,10]. The reasons are plentiful: fluorescent organic dyes are not only 20-fold smaller (~0.5–2 kDa compared to 25 kDa for GFP) but they can also offer superior photophysical properties and more spectral variants compared to FPs. Consequently, a number of methods for the attachment of fluorescent dyes to proteins of interest have been developed, such as widely used Halo[11], SNAP[12], and CLIP tags[13]. However, similarly to genetically encoded FPs, these labeling approaches require making protein fusions, which can affect the function of the protein of interest (POI). This is a general problem that also applies to neuroscience studies involving neuronal cytoskeletal elements, ion channels, receptors, and other synaptic proteins[14].

As an alternative, site-specific labeling utilizing unnatural amino acids (UAAs, also referred to as noncanonical or nonnatural amino acids) and bioorthogonal click chemistry is emerging as a powerful approach for directly labeling proteins with small organic fluorophores[15,16]. These reactions are perhaps best exemplified by the strain-promoted inverse electron-demand Diels–Alder cycloaddition (SPIEDAC) between a strained alkene/alkyne and a tetrazine[17,18]. Due to its extremely high reactivity (reaction rates $>10^5 M^{-1} s^{-1}$) and live-cell compatibility, SPIEDAC is a perfect candidate for the fast, specific, and efficient labeling of biomolecules. To exploit these advantages for protein labeling in living systems, UAAs carrying click-chemistry-reactive moieties in their side chains ("clickable" UAAs) need to be incorporated into target proteins. This can be achieved by amber codon suppression, an emerging area of genetic code expansion[19,20] utilizing orthogonal translational machinery to direct co-translational and site-specific UAA incorporation into the POI (Fig. 1a). SPIEDAC-reactive UAAs, such as TCO*A-Lys (1, Fig. 1b), contain strained alkyne or alkene moieties in their side chains. They can be genetically encoded in mammalian cells with the help of the naturally occurring pyrrolysine (Pyl)-specific amber codon suppressor tRNA[Pyl] and its cognate aminoacyl-tRNA synthetase (PylRS) from *Methanosarcina* species (*M. barkeri* and *M. mazei*), the binding pocket of which has been modified to accommodate the side chains of clickable UAAs[21–24]. SPIEDAC reaction between genetically encoded clickable UAAs and tetrazine derivatives of fluorescent dyes combines a fast, bioorthogonal reaction with site-specific labeling (Fig. 1a, b). This type of labeling provides several advantages, the most notable being the opportunity to attach the fluorescent dye with single-residue precision directly to the POI. This reduces the potential negative impact of the fluorescent tag on the function of the POI

and has steric advantages in the context of super-resolution imaging, as is discussed below.

Since pioneering studies were published demonstrating its suitability for conventional imaging of epidermal growth factor receptor[21] and super-resolution imaging of insulin receptor and influenza virus hemagglutinin protein[25], SPIEDAC-based labeling has emerged as a powerful approach for protein labeling in living mammalian cells. In the meantime, amber codon suppression using a conventional PylRS/tRNA[Pyl] pair, a more efficient PylRS variant fused to a nuclear export signal (NES PylRS)[26], as well as rationally designed tRNA[Pyl] variants[27], has been applied for the fluorescent labeling and imaging of receptors, cytoskeletal proteins, nucleoporins, viral proteins, and other protein structures, as recently reviewed[28]. However, all of these studies were performed in readily transfected cell lines, and this type of labeling has never been used for microscopy studies in complex cells, such as primary neurons.

Here, by using neurofilament light chain (NFL) as a target protein, we established UAA-based SPIEDAC-labeling in living neurons. NFL is the smallest of the three major neurofilament subunits. It interacts with neurofilament medium and neurofilament heavy chain, as well as with α-internexin (in the central nervous system) and peripherin (in the peripheral nervous system) to form an extensive network of neurofilaments. Neurofilaments are intermediate filaments with a 10 nm diameter that have important roles in regulating axonal diameter and conduction velocity, and in modulating synaptic plasticity and activity[29,30]. Until now, fluorescent labeling of neurofilament subunits has been achieved by making FP fusions. However, fusing an FP to a neurofilament subunit might interfere with neurofilament assembly[31]. Here, we describe an alternative, minimally invasive method for labeling NFL in living neurons. The versatility of our labeling approach allowed us to use it for fixed-cell, live-cell, dual-color pulse–chase, and super-resolution imaging of NFL. We also show that amber codon suppression and SPIEDAC labeling can be combined with CRISPR/Cas9 genome engineering for labeling endogenous neuronal proteins.

## Results

**Establishing and optimizing genetic code expansion and click labeling in a rodent ND7/23 neuroblastoma cell line.** To establish site-specific SPIEDAC-based fluorescent labeling of NFL (hereafter also referred to as "click labeling") in primary neurons, we needed to identify optimal transfection conditions for wild-type NFL (NFL[WT]) and its amber mutants (NFL[TAG]). Proper expression of NFL[WT] is a prerequisite for optimization of the subsequent click labeling step, which involved testing several NFL[TAG] mutants and selecting the one that performed best in terms of expression and labeling efficiency. Because the transfection efficiency in primary neurons is low, we first wanted to identify an intermediate host cell line for the amber codon suppression of NFL[TAG] mutants. For this purpose, we tested ND7/23 cells, a neuron-derived immortalized cell line. NFL[WT] expresses well in ND7/23 cells (Fig. 1c) and forms a neurofilament network when co-transfected with a neurofilament medium chain (NFM).

To establish the click labeling of NFL[TAG] in ND7/23 cells, we initially generated four different amber mutants: NFL[K211TAG], NFL[K363TAG], NFL[R438TAG], and NFL[K468TAG]. When selecting the positions of the amber mutations, we avoided all residues known to undergo post-translational modifications, as well as all those involved in the pathology of NFL-associated diseases. The creation of multiple site-specific TAG mutants is an essential step in establishing a protocol for click labeling a new target protein[32]. The reason for this is inherent to the amber codon suppression technology, the efficiency of which depends on the

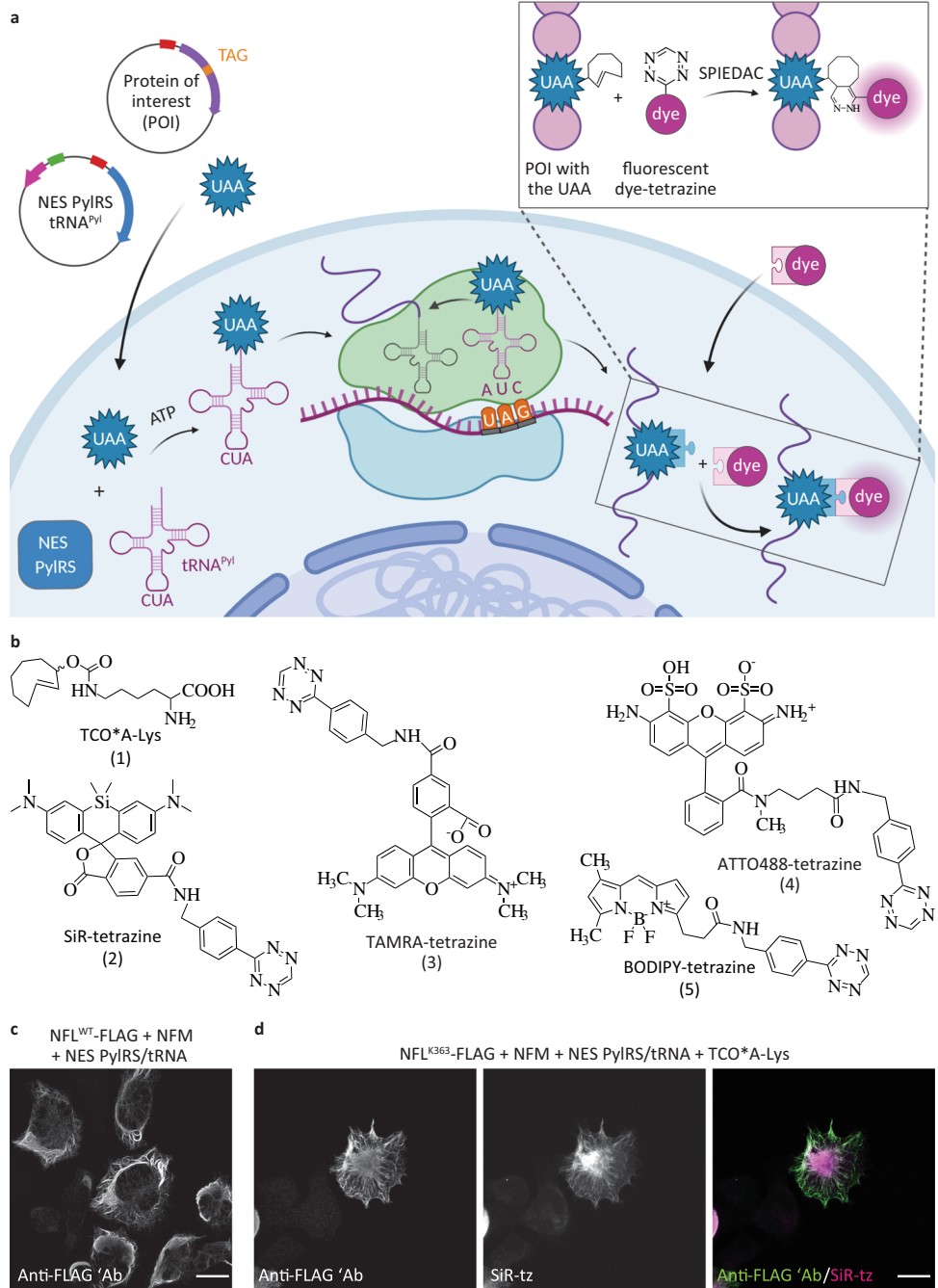

**Fig. 1 Genetic code expansion and click labeling of neurofilament light chain (NFL) in ND7/23 neuroblastoma cells. a** A schematic representation of genetic code expansion and labeling of proteins by click chemistry. Cells are transfected with plasmids bearing genes that encode for NES PylRS (a pyrrolysyl-tRNA synthetase with a nuclear export signal) and $tRNA_{CUA}^{Pyl}$, as well as a TAG stop codon-containing gene that encodes for the protein of interest (POI). The NES PylRS charges $tRNA_{CUA}^{Pyl}$ with the unnatural amino acid (UAA), this charged $tRNA_{CUA}^{Pyl}$ recognizes the UAG stop codon in the mRNA encoding the POI, and UAA is co-translationally incorporated into the POI. In the subsequent step, a tetrazine derivative of a fluorescent dye reacts with the UAA by strain-promoted inverse electron-demand Diels–Alder cycloaddition (SPIEDAC) reaction, and the dye is attached directly to the POI. **b** Chemical structures of UAAs and tetrazine dyes that were used to obtain the data shown in the main figures. **c** ND7/23 cells expressing neurofilament medium-chain (NFM), NES PylRS/$tRNA_{CUA}^{Pyl}$, and wild-type $NFL^{WT}$-FLAG, stained with an anti-FLAG antibody, followed by Alexa Fluor (AF) 488-conjugated secondary antibody. **d** ND7/23 cells expressing $NFL^{K363TAG}$-FLAG, NFM, and NES PylRS/$tRNA_{CUA}^{Pyl}$. Cells were incubated overnight with TCO*A-Lys and then click labeled with silicon rhodamine-tetrazine (SiR-tz). Afterward, cells were fixed and stained with the anti-FLAG antibody, followed by AF488-conjugated secondary antibody. Z-stack images were acquired with a confocal scanning microscope, and are shown as maximum intensity projections. Scale bars: 20 µm (**c**, **d**). Data were collected from three independent experiments (**c**, **d**). Scheme in panel **a** was partially created with BioRender.com.

TAG position and its nucleotide sequence context. In addition, the chosen TAG position needs to be specifically suppressed only in the presence of the UAA and should be accessible to a tetrazine dye for efficient SPIEDAC labeling. We initially tested silicon rhodamine (SiR)-tetrazine (2, Fig. 1b), a cell-permeable fluorophore that is widely used in live-cell and super-resolution microscopy studies[33]. Widefield microscopy revealed that the NFL$^{TAG}$ mutants varied in their expression level and labeling efficiency when cells were co-transfected with the NESPylRS/tRNA$^{Pyl}$ plasmid in the presence of the TCO*A-Lys (Supplementary Fig. 1). Furthermore, screening of multiple mutants allowed us to identify and exclude those that show UAA-independent readthrough of the TAG codon. For example, NFL$^{K211TAG}$ stained positive for an anti-FLAG signal in the absence of the UAA, suggesting synthesis of the full-length protein (Supplementary Fig. 1a); however, as was to be expected, labeling with SiR-tetrazine was unsuccessful. The mutant NFL$^{R438TAG}$ was unable to form a normal neurofilament network and aggregated in the cytoplasm (Supplementary Fig. 1c). Of the two other mutants, NFL$^{K363TAG}$ was expressed at higher levels and was more efficiently labeled with SiR-tetrazine (Fig. 1d and Supplementary Fig. 1b, d). The differences between NFL$^{TAG}$-FLAG mutant expression levels were confirmed by western blot analysis using anti-FLAG antibody (Supplementary Fig. 2). Among the four tested TAG mutants, the efficiency of the amber codon suppression for NFL$^{K363TAG}$ mutant was the highest, as its expression levels corresponded to around 40% of the levels of NFL$^{WT}$ (Supplementary Fig. 2a–c and Supplementary Table 1). As expected, expression levels of TAG mutants were higher in the presence of the mutant eukaryotic release factor 1 (eRF1$^{E55D}$) which was previously used to increase the efficiency of amber codon suppression[34]. Along with the expression levels, we also assessed the amber codon suppression efficiency of our four TAG mutants. By placing the FLAG tag at the N terminus, anti-FLAG western blot allowed us to measure not only the amount of the full-length, but also truncated FLAG-NFL proteins (Supplementary Fig. 2d–f and Supplementary Table 2). As expected for amber codon suppression technology, different TAG positions led to different amounts of truncated fragments, but in general, the expression of the full-length NFL was higher and the amount of truncated NFL was lower in the presence of the mutant eRF1 factor (Supplementary Fig. 2d–f and Supplementary Table 2). Interestingly, mutant NFL$^{R438TAG}$ that failed to form normal neurofilament network (Supplementary Fig. 1c) showed a very high level of expression of the truncated fragment suggesting that this truncated fragment might be affecting neurofilament assembly. For our most efficient NFL$^{K363TAG}$ construct, the amount of truncated fragment relative to the amount of total expressed NFL$^{K363TAG}$ was on average corresponding to 20% (Supplementary Fig. 2f and Supplementary Table 2). Although truncated NFL could potentially affect the assembly of neurofilaments, in the case of NFL$^{K363TAG}$ construct we saw no effect on cell viability or neurofilament network formation. Furthermore, average amount of truncated protein detected on western blot does not mean that each individual cell will express 20% of the truncated fragment. Finally, to make this method even more robust and to avoid any potential risks related to the unknown effects of truncated proteins, we aimed to lower their amount even further. To do this, we tested additional, improved constructs for amber codon suppression. When using plasmids encoding codon-optimized NES PylRS or multiple copies of designer tRNA[27], the amount of truncated fragment was reduced to as little as 2.6% and 10% of the total expressed NFL$^{K363TAG}$, respectively (Supplementary Fig. 3).

Because of its high expression levels and labeling efficiency, we subsequently used NFL$^{K363TAG}$ mutant to compare different

UAAs and other tetrazine dyes. Since the click reaction product of TCO*A-Lys and tetrazines can undergo elimination reaction which in turn would lead to the release of the fluorophore from the protein target and nonspecific fluorescence background[35–38], we wanted to test additional UAAs, such as endo-BCN-Lys (7, Supplementary Fig. 4) and TCO4en/eq-Lys (6, Supplementary Fig. 4). The former has been previously successfully used for intracellular protein labeling[39,40] and the latter has been recently rediscovered as a very stable UAA for click labeling[41]. Experiments in ND7/23 cells show that both can be genetically encoded into NFL$^{K363TAG}$ and specifically labeled with SiR-tetrazine (Supplementary Fig. 4). To take advantage of the broad palette of available tetrazine-dye derivatives, including cell-impermeable variants, we tested additional dyes for click labeling of NFL$^{K363TAG}$ in living and fixed ND7/23 cells (Supplementary Fig. 5). Although click labeling of NFL$^{K363TAG}$ was possible with both cell-permeable (SiR-tetrazine, TAMRA-tetrazine (3, Fig. 1b), BODIPY-tetrazine (5, Fig. 1b), Janelia Fluor (JF) 646-methyl-tetrazine, CF650-methyl-tetrazine, JF549-tetrazine, CF500-methyl-tetrazine,) and cell-impermeable tetrazine dyes (ATTO488-tetrazine (4, Fig. 1b), Alexa Fluor 647-tetrazine, ATTO655-methyl-tetrazine), we noted that the cell-impermeable dyes gave higher background fluorescence, which was most likely caused by the fact that the labeling was done post-fixation and not in living cells. Taken together, these results show that ND7/23 neuroblastoma cells are a good host for the genetic code expansion and click labeling. They are transfected with high efficiency and allowed us to identify the most suitable NFL$^{TAG}$ mutant and to optimize SPIEDAC labeling conditions.

**Expression and click labeling of NFL$^{K363TAG}$ in living primary mouse neurons**. After identifying the optimal amber mutant position and conditions for click labeling of NFL$^{TAG}$ in ND7/23 cells, we focused on experiments in primary mouse cortical neurons (MCNs). Analogously to the experiments in ND7/23 cells, we tested three different UAAs: TCO*A-Lys, endo-BCN-Lys, and TCO4en/eq-Lys. When co-transfected with NESPylRS/tRNA$^{Pyl}$ in the presence of TCO*A-Lys, the NFL$^{K363TAG}$ mutant was expressed well and assembled into neurofilament network in primary neurons (Fig. 2a). This is very important because the capacity to assemble into neurofilament network is the sole criterion used in the literature to assess the suitability of neurofilament constructs for microscopy and cell biology studies[31]. Furthermore, SiR-tetrazine signal co-localized with anti-FLAG (although different primary antibodies showed different levels of colocalization) and anti-NFL antibodies indicating that specific click labeling of NFL$^{K363TAG→TCO*A-Lys}$ had occurred (Fig. 2a, b and Supplementary Fig. 6).

In contrast to the results obtained with ND7/23 cells, TCO4en/eq-Lys was not efficiently incorporated in NFL$^{K363TAG}$ in primary neurons. This is probably related to the fact that the PylRS$^{AF}$ that we used poorly accepts TCO4en/eq-Lys[41]. Consequently, we found only a few neurons with fully translated and click-labeled NFL$^{K363TAG→TCO4en/eq-Lys}$. On the other hand, incorporation of endo-BCN-Lys during the 3-day incubation period induced cytotoxicity. We do not fully understand why this happens, but as our experimental protocols involved longer incubation periods, we continued using TCO*A-Lys in subsequent experiments.

In addition to SiR-tetrazine, we evaluated the suitability of other cell-permeable and cell-impermeable dyes for click labeling of NFL$^{K363TAG→TCO*A-Lys}$ in living and fixed neurons, respectively (Supplementary Fig. 7). NFL$^{K363TAG→UAA}$ showed higher expression levels in neurons than in ND7/23 cells, and

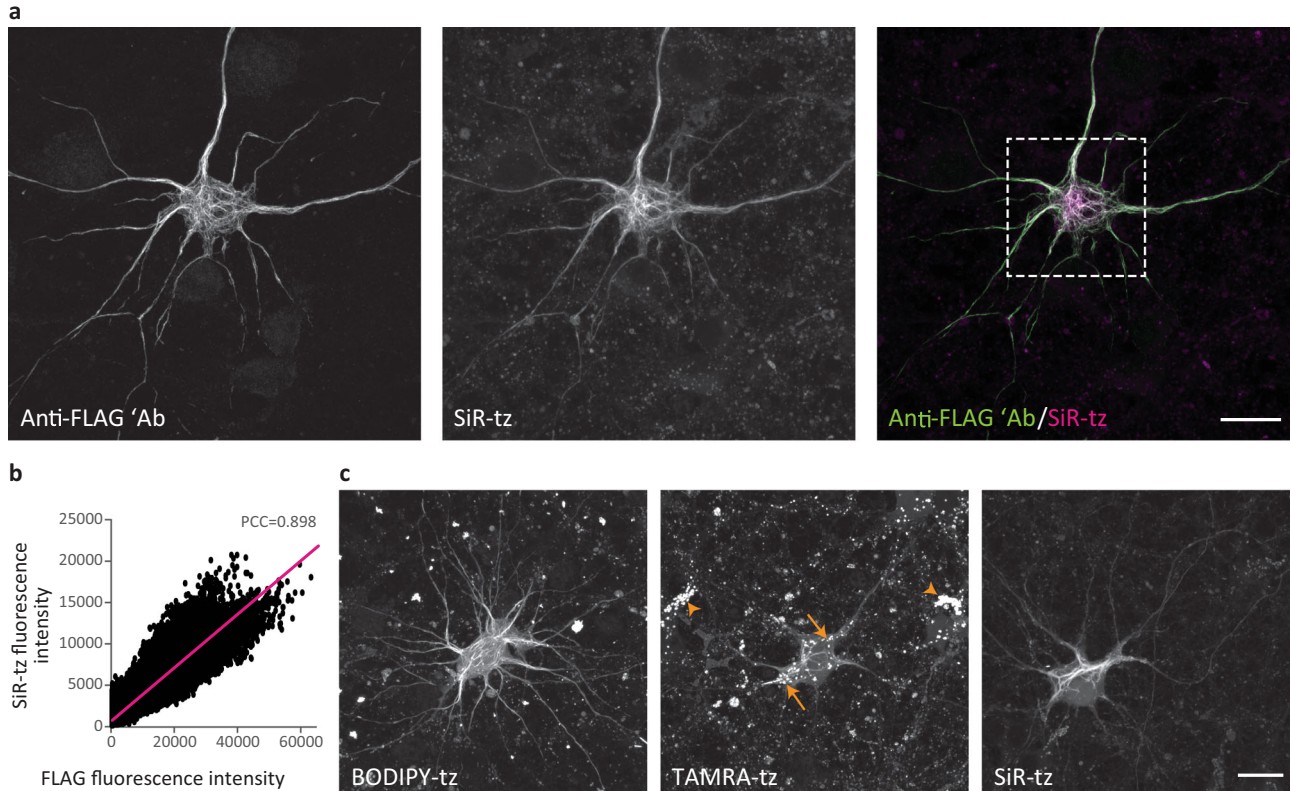

**Fig. 2 Genetic code expansion and click labeling of NFL in live primary mouse cortical neurons (MCNs). a**, **c** MCNs expressing NFL^K363TAG-FLAG, NFM and NES PylRS/tRNA_CUA^Pyl in the presence of TCO*A-Lys. **a** Three days after transfection, neurons were labeled with SiR-tz via click chemistry, then fixed and stained with anti-FLAG antibody, followed by AF488-conjugated secondary antibody. After fixation and immunocytochemistry labeling, neurons were imaged on a confocal scanning microscope. **b** Colocalization analysis of SiR-tz and FLAG intensity in the region within the dashed box from panel **a**, with Pearson's correlation coefficient (PCC) shown above the scatterplot. Fluorescence intensities in the graph are shown as absolute gray values of 16-bit depth images. **c** Three days after transfection, neurons were labeled with either BODIPY-tz, TAMRA-tz, or SiR-tz. Excess dye was washed for 3 h, and live neurons were imaged on a confocal scanning microscope. In TAMRA-tz panel, background staining of lysosomes is highlighted in both NFL^K363TAG_-expressing neuron (arrows) and in non-transfected neurons (arrowheads). Z-stack images are shown as maximum intensity projections. Scale bars: 20 μm (**a**, **c**). Data were collected from three independent experiments. Source data are provided as a Source Data file.

consequently higher intensity of click-labeled NFL. This is exemplified by JF646-methyl-tetrazine, which gave a weak signal in ND7/23 cells after the labeling step compared to that observed in primary neurons (Supplementary Figs. 5b and 7b). The possibility of combining amber codon suppression and SPIEDAC in primary neurons encouraged us to further explore the suitability of this labeling approach in a range of microscopy studies.

**Click labeling allows imaging of NFL in living neurons.** Since a number of cell-permeable dyes showed specific fluorescent labeling of NFL^K363TAG→UAA (Supplementary Fig. 7a–g), we tested the compatibility of our labeling approach with live-cell imaging. We successfully imaged living neurons expressing NFL^K363TAG→TCO*A-Lys that we labeled with one of the cell-permeable dyes, including tetrazine derivatives of BODIPY, TAMRA, and SiR (Fig. 2c). A comparison of labeling quality in live-imaging experiments revealed that different dyes emit different background levels, with SiR- and BODIPY-conjugated tetrazines affording a higher signal-to-noise ratio than TAMRA-tetrazine. In addition, we observed a nonspecific signal in cytoplasmic vesicles during live-cell imaging. This type of background was not present in images of fixed neurons (Fig. 2a). Control experiments with SiR-tetrazine revealed that lysosomes accumulated UAA molecules that were labeled with tetrazine dyes

independently of the amber codon suppression (Supplementary Fig. 8). This was confirmed by incubating non-transfected neurons with UAA and SiR-tetrazine dye (Supplementary Fig. 8c). Although SiR-tetrazine has some residual affinity for lysosomes in the absence of a UAA (Supplementary Fig. 8b,d), the majority of nonspecific lysosomal labeling seems to occur due to the accumulation of UAA in lysosomes (Supplementary Fig. 8c). Additional control experiments excluded the possibility of tetrazine dyes reacting with lysotracker, as both SiR-tetrazine and BODIPY-tetrazine showed the same lysosomal accumulation independently of lysotracker labeling (Supplementary Fig. 9). This type of nonspecific labeling is important to keep in mind for future studies, especially if lysosomal proteins are to be labeled, although interestingly, lysosomes are labeled with lower intensity in neurons that express higher levels of NFL^TAG and form more extensive neurofilament networks. This can be seen by comparing lower-expressing TAMRA-tetrazine-labeled with higher-expressing SiR- or BODIPY-tetrazine-labeled neurons (Fig. 2c). In addition, this lysosome-associated background is less apparent after fixation and cell permeabilization (Fig. 2a).

We also investigated sources of the diffuse cytoplasmic background after intracellular click labeling in neurons. To quantify this type of background, we focused on two cell-permeable dyes, SiR- and BODIPY-tetrazine, which performed well in both live-cell and fixed-cell imaging experiments. Additionally, we wanted to compare background levels with

TCO*A-Lys to those with endo-BCN-Lys, since the latter should give less background as its click product does not undergo elimination and BCN-Lys is more easily washed out of the cell[42]. Thus, we shortened the UAA incubation period from 3 to 2 days, to avoid cytotoxicity previously observed during longer incubation with endo-BCN-Lys. We quantified the intensity of cytoplasmic fluorescence background in NFL$^{K363TAG \rightarrow UAA}$ expressing neurons, that were labeled live with SiR- or BODIPY-tetrazine and fixed at two time points (2 h or 10 h) after the labeling. Our results show the same trend with both dyes – when compared at 2 h after the click reaction, TCO*A-Lys gives higher background than endo-BCN-Lys (Supplementary Fig. 10). Additionally, the TCO*A-Lys background can be reduced to the level of endo-BCN-Lys background by prolonged washing steps (Supplementary Fig. 10). Although we have not quantified it here, the same can probably be achieved by titrating the concentration of the UAA and further adjustments of the UAA incubation or tetrazine-dye washing step durations. These results are in line with the published literature and suggest that endo-BCN-Lys might be a better choice for intracellular click labeling. However, as cell viability and health are crucial for the experiments involving primary neurons, and all subsequent experiments included incubation periods longer than 3 days, we continued using TCO*A-Lys for the expression of NFL$^{K363TAG}$ in neurons.

**Dual-color pulse–chase click labeling of NFL in primary neurons.** One advantage of UAA-based SPIEDAC labeling is the flexibility that it offers with regard to the choice of the tetrazine dye. As shown above, different cell-permeable and cell-impermeable dyes could be used for live-cell and fixed-cell labeling, allowing the possibility of multi-color labeling. We wanted to take advantage of this in order to achieve dual-color labeling in a time-dependent (pulse–chase) manner, as outlined in Fig. 3a. By incubating transfected neurons with two tetrazine dyes at two distinct time points (10 vs. 12 days in vitro), we labeled different populations of NFL$^{K363TAG}$, which were synthesized during defined phases of neuronal growth in vitro. For these experiments, we combined a pair of cell-permeable dyes. We established a dual-color labeling protocol with a combination of BODIPY-tetrazine and SiR-tetrazine, which allowed us to image two populations of NFL in fixed (Fig. 3b and Supplementary Fig. 11a) and living neurons (Fig. 3c). For this type of pulse–chase labeling, it is essential that the first dye labels all NFL$^{TAG}$ molecules that have been expressed up to the point the dye is applied. If this happens, then the second dye will label the NFL molecules that were synthesized subsequently. To test this, we performed control experiments in which the second dye showed no labeling when applied immediately after the first dye or after 2 days of incubation in the absence of UAA (Supplementary Fig. 11b, c). Although conventional confocal microscopy imaging (Fig. 3b, c and Supplementary Fig. 11a) suggests that most of the neurofilaments in neuronal cell bodies overlap, the potential of this labeling approach lies with studies of target protein trafficking during neuronal growth, injury, or other biological processes. To illustrate this better, we changed the time course of the pulse−chase labeling procedure. Firstly, we tried to label newly synthetized NFL by performing the second click reaction 3 h instead of 2 days after the first click reaction. Confocal microscopy revealed that newly synthetized NFL gets incorporated into the existing network relatively fast since most of the filaments overlapped in two channels (Fig. 3d). In addition, we could identify cases of two-color chimeric filaments (Fig. 3d inset) suggestive of end-to-end annealing of filaments as described in the literature[43]. To explore what happens with newly synthetized NFL during longer periods of time, we changed the

labeling scheme again. In this case, after click labeling of the first NFL$^{K363TAG}$ population, we introduced an additional step during which no UAA was added to the medium, and the translation of the second NFL$^{K363TAG}$ population was paused (Fig. 4a). This modification of the protocol resulted in better-defined and spatially separated NFL populations in comparison to the original procedure shown in Fig. 3. NFL that was synthetized earlier during neuronal growth and labeled with the first click reaction was mainly transported distally along axons, while newly synthetized NFL was mainly present in the cell bodies (Fig. 4b). Depending on the expression levels, this difference was more or less obvious at the level of individual neurons.

**Click labeling with minimal fluorescent tags allows super-resolution imaging of NFL in primary neurons.** An optimal labeling tag is small and minimizes the impact on the protein of interest. Furthermore, the minimal size of a UAA-based tag, and the possibility to attach the dye directly to the target protein, make them particularly attractive for super-resolution imaging. To test if SPIEDAC labeling of NFL in primary neurons is compatible with super-resolution imaging, we performed stimulated emission depletion (STED) imaging of SiR-tetrazine-labeled NFL$^{K363TAG}$. This allowed us to resolve NFL bundles with improved resolution compared to confocal imaging (Fig. 5a–d). In addition, we imaged NFL$^{K363TAG}$ labeled with ATTO488-tetrazine and CF650-methyl-tetrazine in neuronal cell bodies (Supplementary Fig. 12a, b) and SiR-tetrazine-labeled NFL$^{K363TAG}$ in neuronal processes (Supplementary Fig. 12c). As STED led to improvement in resolution, we combined our pulse–chase dual-color labeling assay with two-color super-resolution imaging. Owing to their compatibility with STED imaging, we used SiR- and ATTO488-tetrazine to resolve two populations of click-labeled NFL in cell bodies (Fig. 5e) and neuronal processes (Supplementary Fig. 13). Since ATTO488-tetrazine is not cell permeable, we combined live- and fixed-cell SPIEDAC labeling. Finally, to further emphasize the potential of our labeling approach and its suitability for advanced microscopy studies, we combined the dual-color pulse–chase labeling with an oxidative-stress axonal injury (Supplementary Fig. 14a). This allowed us to reveal the distribution of different NFL populations with nanoscale resolution in healthy and injured axons with STED microscopy (Supplementary Fig. 14b, c).

**CRISPR/Cas9 genome editing can be combined with genetic code expansion and SPIEDAC chemistry to achieve labeling of endogenous proteins.** To date, click labeling has been achieved by transfecting the POI$^{TAG}$ and expressing it under strong promoters, such as CMV. Although working with transfected cells is routine for cell biologists, overexpression of the POI can affect its function and be toxic for the host cell. We tried to overcome this limitation by combining click labeling and CRISPR/Cas9 genome editing. We used the CRISPR/Cas9-based knock-in strategy based on the recently published ORANGE (Open Resource for the Application of Neuronal Genome Editing) toolbox[44]. As a proof-of-concept, we first mutated the previously described pORANGE vector for tagging endogenous βIII-tubulin with GFP. We introduced a Y39TAG mutation in the GFP gene and showed that the full-length βIII-tubulin-GFP$^{Y39TAG}$ was only synthesized in the presence of NESPylRS/tRNA$^{Pyl}$ and UAA (Supplementary Fig. 15a, b). This demonstrated that amber codon suppression and CRISPR/Cas9 could be combined. We then designed a vector for the tagging of endogenous NFL (Fig. 6a). Because of the limited efficiency of amber codon suppression, site-specific introduction of TAG at the position K363 would result in knockdown of the endogenous NFL. Therefore, we changed the

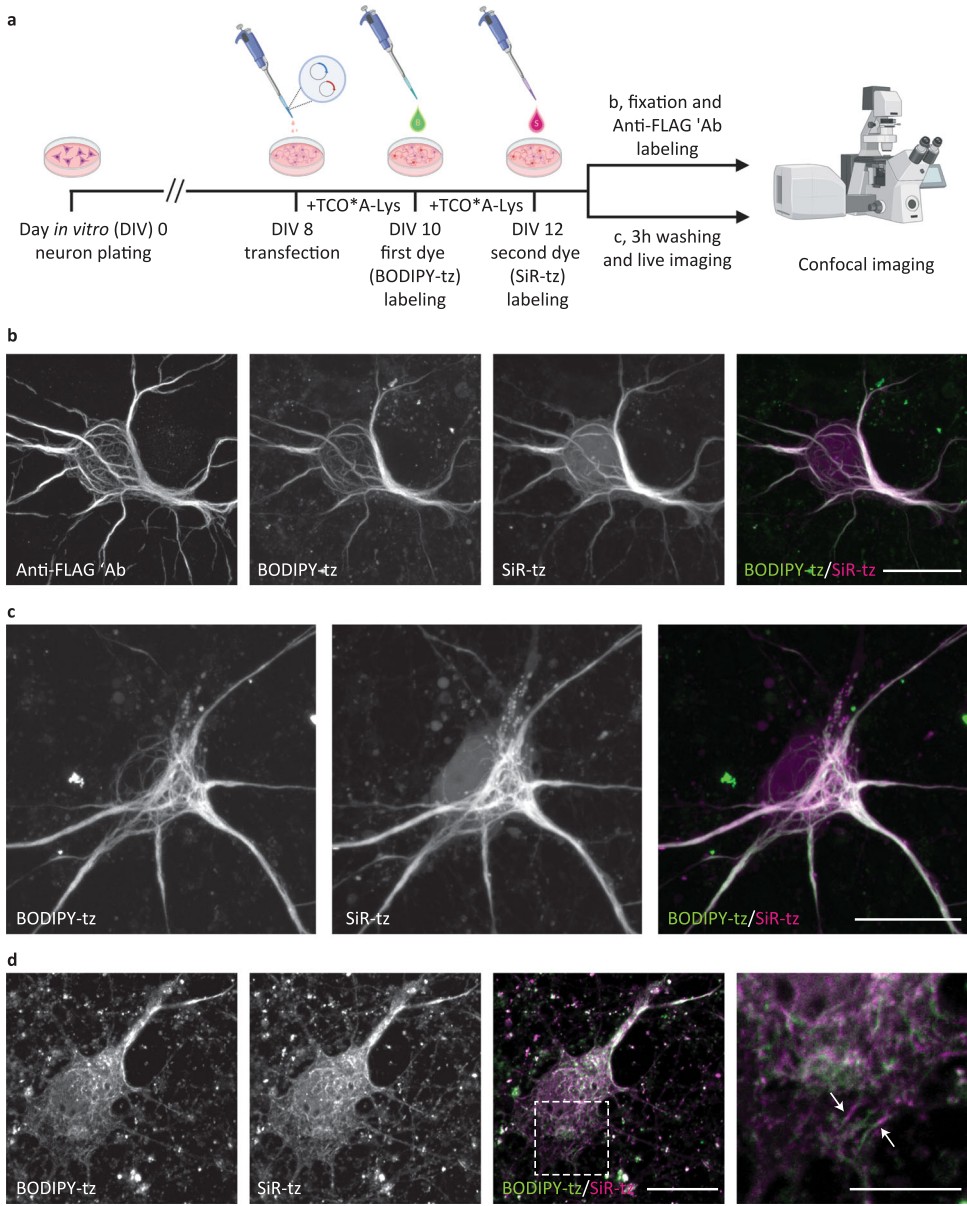

**Fig. 3 Pulse–chase click labeling of two NFL populations in live primary mouse cortical neurons (MCNs). a** A schematic representation of the experimental workflow. Eight days after plating, MCNs were transfected with NFL$^{K363TAG}$-FLAG, NFM, and NES PylRS/tRNA$_{CUA}^{Pyl}$ constructs. After 2 days of incubation with TCO*A-Lys, neurons were labeled with the first dye (BODIPY-tz), incubated with TCO*A-Lys for a further 2 days, and labeled with the second dye (SiR-tz). After the second labeling step, neurons were either fixed, stained with anti-FLAG antibody followed by AF555-conjugated secondary antibody, and imaged on a confocal scanning microscope (**b**), or live neurons were imaged on a confocal scanning microscope (**c**). Z-stack images are shown as maximum intensity projections. **d** MCNs expressing NFL$^{K363TAG}$-FLAG, NFM, and NES PylRS/tRNA$_{CUA}^{Pyl}$, labeled with a modified dual-color labeling approach. MCNs were transfected and labeled with the first tetrazine dye in the same way as in **a–c**. After the first dye labeling, neurons were incubated with TCO*A-Lys for 3 h, labeled with the second dye, fixed, and immunostained. Single-plane images were acquired on a confocal scanning microscope. Arrows indicate chimeric neurofilaments composed of SiR-tz and BODIPY-tz labeled NFL segments. Scale bars: 20 μm (**b–d**), 10 μm (inset in **d**). Data were collected from three independent experiments. Scheme in panel **a** was partially created with BioRender.com.

approach and instead of introducing a site-specific TAG codon at the position K363, we added a hexapeptide linker followed by a 3xFLAG tag at the C terminus of NFL (Fig. 6b). For the purpose of amber codon suppression and click labeling, we introduced a TAG codon at position A6 of the linker (linker$^{A6TAG}$; Fig. 6c). Tagged endogenous NFL can be detected with anti-FLAG staining (Fig. 6d, e). Co-staining with the anti-NFL antibody confirmed the accuracy of the knock-in, as the anti-FLAG signal overlapped with that of anti-NFL (Fig. 6d). The specificity was also confirmed by sequencing of the PCR product flanking the integration sites in genomic DNA isolated from transfected

neurons (Supplementary Fig. 15c, d). As expected, sequencing results showed a mixture of genomic DNA sequences resulting from neurons with both successful and unsuccessful knock-in. For example, our unsuccessful knock-in contained a single nucleotide insertion. These types of insertions are a common consequence of genome editing and would lead to translational frame shifts. However, the fact that immunocytochemical labeling with anti-FLAG antibody detected correctly translated NFL-FLAG confirms that this insertion is not present in all genomic sequences. In addition, since the only suitable protospacer adjacent motif (PAM) site at the end of the *Nefl* gene was located

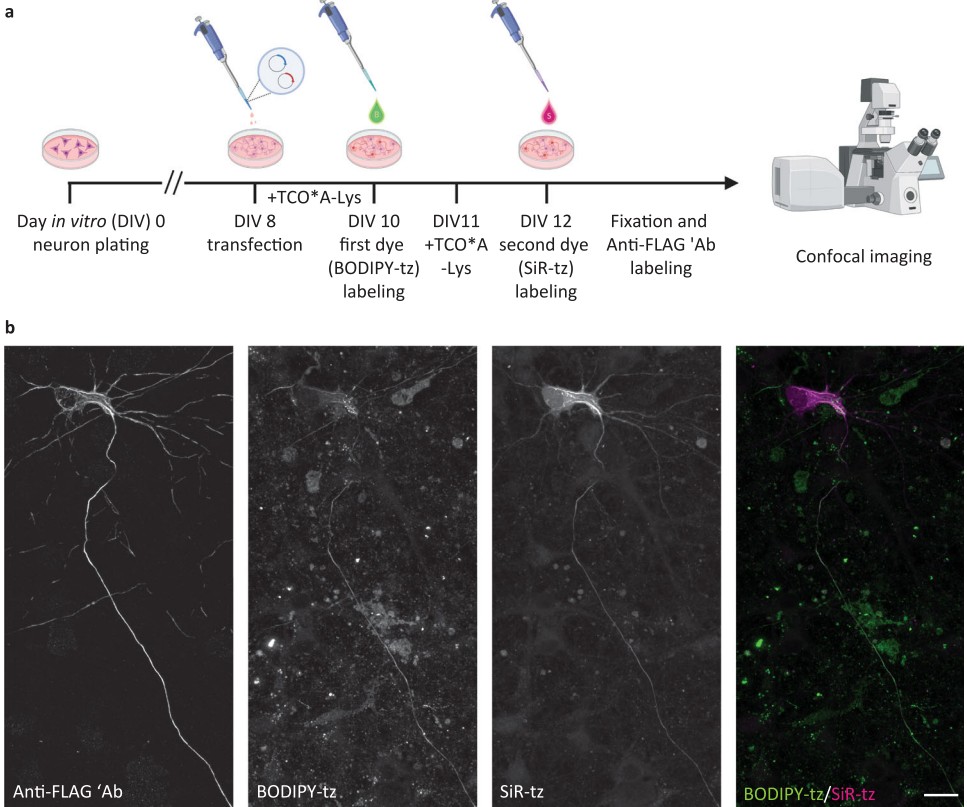

**Fig. 4 Modified pulse–chase click labeling approach reveals two spatially separated NFL populations in cortical neurons (MCNs). a** A schematic representation of the modified experimental workflow. Eight days after plating, MCNs were transfected with NFL$^{K363TAG}$-FLAG, NFM, and NES PylRS/tRNA$_{CUA}^{Pyl}$ constructs. After 2 days of incubation with TCO*A-Lys, neurons were labeled with the first dye (BODIPY-tz), then incubated without UAA for 1 day. Afterward, neurons were incubated with TCO*A-Lys for another day, and labeled with the second dye (SiR-tz). After the second labeling step, neurons were fixed, stained with anti-FLAG antibody followed by AF555-conjugated secondary antibody, and imaged on a confocal scanning microscope. **b** Representative Z-stack images showing the two NFL populations. Scale bar: 20 µm (**b**). Data were collected from three independent experiments. Scheme in panel **a** was partially created with BioRender.com.

upstream of the stop codon, ORANGE-based knock-in resulted in the deletion of six C-terminal amino acids. This was also confirmed by sequencing. The efficiency of the knock-in 6 days after transfection was around 20% (Supplementary Fig. 15e). For click labeling, neurons were transfected with pORANGE NFL linker$^{A6TAG}$-3xFLAG knock-in (KI) construct, together with the NESPylRS/tRNA$^{Pyl}$ and mutant eRF1$^{E55D}$ to increase the efficiency of amber codon suppression (Supplementary Figs. 2 and 3). In the presence of TCO*A-Lys, the UAG codon in the linker is suppressed, and endogenous NFL is tagged with linker-$^{A6TAG \rightarrow TCO*A-Lys}$-3xFLAG and can be labeled by click chemistry (Fig. 6e). In the absence of TCO*A-Lys, translation finishes at the UAG codon, and the full-length NFL is translated, leaving endogenous NFL levels unaffected (Fig. 6c). These results show that it is possible to combine genetic code expansion and click labeling with CRISPR/Cas9 genome editing for the labeling of endogenous NFL. However, depending on the endogenous expression levels of the protein of interest, the specific click labeling could be masked by the readthrough of endogenous amber codons. The suppression of endogenous amber codons with orthogonal tRNA would lead to the synthesis of C-terminally extended proteins. While we expect most of such C-terminally extended proteins to get degraded[45], they could potentially get click labeled and contribute to the measured fluorescence signal. While this is not obvious in overexpression experiments, it could become a problem when labeling proteins expressed under endogenous promoters. To control for this, we transfected MCNs with wild-type pORANGE NFL linker-

3xFLAG KI and performed click labeling with SiR-tetrazine, following the same protocol used for the labeling of the clickable NFL-linker$^{A6TAG}$-3xFLAG mutant. Click labeling signal, although present, was lower in this case (Supplementary Fig. 16a). However, this background signal varied greatly among transfected cells, depending on the transfection efficiency and levels of expression. Since eRF1$^{E55D}$ itself could contribute to this background due to increased amber codon suppression not only of our site-specifically introduced amber codon but also native amber codons, we performed an additional control experiment. We transfected MCNs with NESPylRS/tRNA$_{CUA}^{Pyl}$, with or without eRF1$^{E55D}$, and performed click-labeling with SiR-tetrazine, in the same way as for knock-in experiments. We included a plasmid encoding for mCherry in order to be able to identify transfected neurons. Microscopy data, as well as quantitative analysis, showed that there is no significant difference in click labeling background of neurons transfected with NESPylRS/tRNA$_{CUA}^{Pyl}$ and eRF1$^{E55D}$ compared to the neurons that were transfected with NESPylRS/tRNA$_{CUA}^{Pyl}$ only (Supplementary Fig. 16b, c). These results suggest that eRF1$^{E55D}$ alone does not contribute significantly to the click labeling background in primary mouse neurons.

As mentioned above, due to the limited number of suitable PAM sites, ORANGE-based editing of *Nefl* gene resulted in the deletion of six C-terminal amino acids. This limitation can be avoided by using alternative genome engineering strategies that rely on targeting of noncoding regions, and which offer more flexibility in choosing the target sites, such as Targeted Knock-In

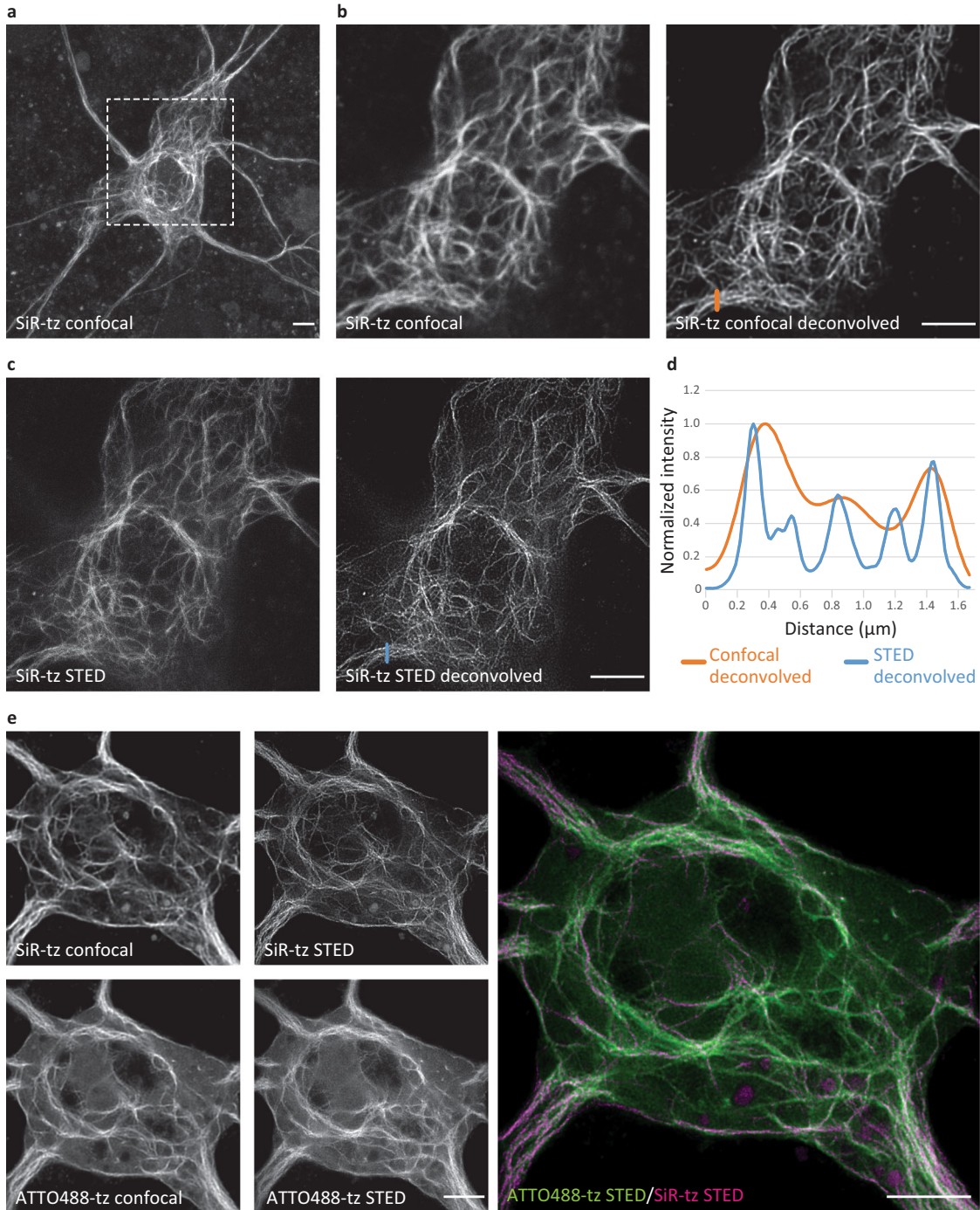

**Fig. 5 Super-resolution stimulated emission depletion (STED) imaging of click-labeled NFL in primary mouse neurons (MCNs).** MCNs expressing NFL$^{K363TAG}$-FLAG, NFM, and NES PylRS/tRNA$_{CUA}^{Pyl}$ in the presence of TCO*A-Lys. **a–c** Two to three days after transfection, neurons were labeled with SiR-tz, fixed, and stained with anti-FLAG antibody. Afterward, neurons were imaged with STED super-resolution microscopy. Panel **a** shows a maximum projection of a confocal Z-stack of the neuron. The next four images show the confocal (**b**) and STED (**c**) micrographs of the region within the dashed box before and after deconvolution. **d** The increase in resolution is visualized in a graphical representation of signal intensities across the line profiles drawn in the deconvolved confocal (orange line) and STED (blue line) images. **e** STED imaging of two populations of click-labeled NFL. Two days after transfection, neurons were labeled with SiR-tz and incubated for a further 2 days with TCO*A-Lys. Afterward, cells were fixed, labeled with ATTO488-tz, and imaged with STED super-resolution microscopy. Raw confocal and STED images were deconvolved using Huygens deconvolution software. Scale bars: 5 μm (**a–c**, **e**). Data were collected from three independent experiments. Source data are provided as a Source Data file.

with Two guides (TKIT) strategy[46]. We applied this technique for tagging of endogenous NFL, and combined it with the genetic code expansion and SPIEDAC labeling, in the same way as described for the ORANGE approach (Supplementary Fig. 17a, b). SPIEDAC labeling of endogenous NFL was successful and the background was similar to the one observed when this labeling was combined with ORANGE knock-in. The efficiency of TKIT knock-in was around 24% after 6 days of expression (Supplementary Fig. 17c). The specificity of the genetic incorporation was confirmed by isolation of total RNA from transfected neurons,

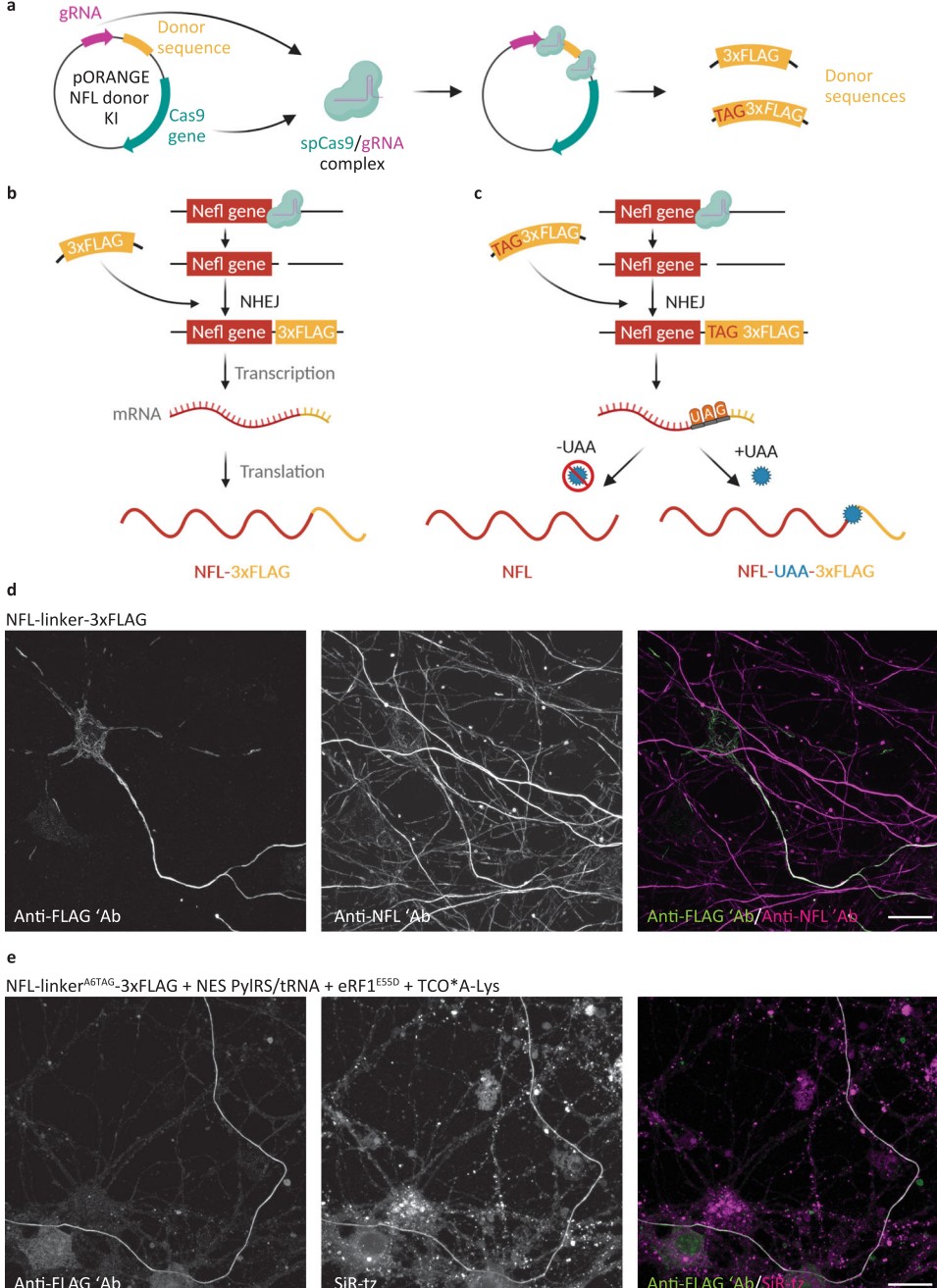

**Fig. 6 Click labeling of endogenous NFL in primary mouse neurons (MCNs). a–c** A schematic representation of the tagging of endogenous NFL. **a** MCNs were transfected with a pORANGE plasmid encoding for spCas9 and gRNA, and containing linker-3xFLAG or linker^A6TAG-3xFLAG donor sequence. SpCas9 and gRNA are expressed from the plasmid and cut target sequences around the donor sequence and at the end of the *Nefl* gene. The resulting donor sequences are used by the non-homologous end joining (NHEJ) system to repair a double-strand break in the genomic DNA. **b** Endogenous NFL bearing the linker-3xFLAG sequence is transcribed and translated with the addition of the linker-3xFLAG tag at the C terminus. **c** Endogenous NFL bearing the linker^A6TAG-3xFLAG tag is fully transcribed. In the absence of a UAA, translation finishes at the UAG codon, and full-length NFL is synthesized. In the presence of a UAA and NES PylRS/tRNA_CUA^Pyl, the UAA is incorporated in response to the UAG codon, the NFL-linker-UAA-3xFLAG is synthesized, and can be click labeled with a tetrazine dye. **d** MCNs transfected with pORANGE NFL linker-3xFLAG knock-in construct, stained with anti-FLAG and anti-NFL antibodies, after 6 days of expression. The anti-FLAG and anti-NFL antibodies were stained with AF488- and AF647-conjugated secondary antibodies, respectively. **e** MCNs transfected with pORANGE NFL linker^A6TAG-3xFLAG knock-in, codon-optimized NES PylRS/tRNA_CUA^Pyl and eukaryotic release factor 1 mutant E55D (eRF1^E55D) constructs. After incubation with TCO*A-Lys for 6 days, the endogenous NFL-linker-UAA-3xFLAG was labeled with SiR-tz, then neurons were fixed and stained with anti-FLAG antibody followed by AF488-conjugated secondary antibody. Scale bars: 20 µm (**d**, **e**). Data were collected from three independent experiments. Schemes in panels **a-c** were partially created with BioRender.com.

cDNA synthesis, and PCR with primers flanking the incorporation sites. Sequencing of PCR products confirmed that the donor DNA is integrated at the correct position and that mRNA splicing of the last two exons was unaffected by the integration (Supplementary Fig. 17d, e). As the editing happens in the intron and 3′ UTR, the main advantage of TKIT approach over ORANGE is the preservation of the intact *Nefl* open reading frame. Together, these results show that two different approaches for genome engineering can be combined with click chemistry labeling of endogenous proteins. Implications and limitations for future research will be discussed in detail below.

## Discussion

Here, we describe minimal tags for the fluorescent labeling of proteins in living primary mouse neurons. Our labeling approach is based on bioorthogonal SPIEDAC click chemistry between site-specifically incorporated UAAs and tetrazine dyes.

Incorporation of UAAs into target proteins by genetic code expansion allows the introduction of novel chemical and physical properties into biological systems. This emerging protein engineering technique opens a plethora of possibilities for studies of proteins in vitro and in vivo, at the single-cell and whole-organism levels[15,16,19,20]. UAAs with diverse side chains have been developed: fluorescent UAAs, clickable UAAs with reactive chemical handles for bioorthogonal labeling, photoresponsive UAAs for light-induced control of protein function, and UAAs for the introduction of post-translational modifications, are some examples. Although in its infancy, genetic code expansion has also been applied in neurobiology, as reviewed recently[47]. In 2007, one of the earliest studies on the genetic encoding of UAAs in mammalian cells pioneered the now widely used system for the expression of orthogonal tRNAs under the type 3 eukaryotic RNA polymerase III promoters in HEK293T cells and primary neurons[48]. The authors used this approach to site-specifically incorporate *O*-methyltyrosine, a UAA with an extended side-chain, to study the inactivation of voltage-gated potassium Kv1.4 channel in HEK293T cells. As a proof-of-principle, they showed incorporation of *O*-methyltyrosine in a reporter GFP[TAG] protein in neurons. In the meantime, other types of UAAs were applied in further studies of neuronal proteins, such as irreversible or reversible light-induced control of NMDA- and AMPA-type glutamate receptors with photoresponsive UAAs[49–52], fluorescent labeling of NMDA receptors[53] and the Shaker B voltage-dependent potassium channel[54], as well as the development of reporters for amyloid precursor protein trafficking and processing[55,56]. However, all of these studies were performed using standard cell lines, and the number of studies showing UAA incorporation in neurons has remained limited. In this respect, the fluorescent UAA dansyl-alanine was used in neuronal stem cells to visualize membrane potential[57]. In another study, photocaged cysteine was used to create a light-induced switch for the inwardly-rectifying potassium channel (KiR2.1) in mouse brain slices and living brain[58]. More recent studies reported the incorporation of diverse UAAs in a reporter GFP[TAG] protein with the help of adeno-associated virus (AAV) and baculovirus vectors in dissociated neurons and organotypic slices[59,60], as well as AAV-based UAA incorporation in mouse brain[59]. We have now expanded the portfolio of genetic code expansion applications in neuroscience by showing that clickable UAAs can be genetically encoded in neurons and used for site-specific fluorescent labeling. During the revision of our manuscript, a complementary work using click chemistry for extracellular labeling of transmembrane AMPAR regulatory proteins in living neurons was published[61], further highlighting the potential of this method. As discussed earlier, this labeling approach has previously only been used in standard and readily transfected cell lines and not in complex cells, such as neurons. Here, we show its versatility by performing advanced microscopy studies involving fixed-cell and live-cell imaging, pulse–chase dual-color labeling, and STED imaging of a click-labeled NFL.

Neurofilaments play important roles in healthy and diseased neurons. Certain alterations in neurofilament genes, such as Charcot–Marie–Tooth disease-associated NFL mutations, are direct causes of neurological diseases. In other neurological diseases, such as amyotrophic lateral sclerosis, and giant axon neuropathy, abnormal transport, and processing of neurofilaments appear to be a secondary consequence of the disease pathology. Furthermore, NFL levels in the cerebro-spinal fluid and blood correlate with the severity of various neurological conditions, and have been proposed as potential biomarkers of disease progression[62]. Nevertheless, our understanding of neurofilament roles in physiological and pathophysiological conditions remains limited. Here, we describe an alternative method for NFL labeling in living neurons, which in combination with the existing methods, might spur novel discoveries in the field.

To facilitate the implementation of click labeling technology for new applications in neurobiology, we provide a workflow for establishing the click labeling of additional target proteins. In this regard, selecting an appropriate host cell line for amber codon suppression is a key prerequisite. As successful amber codon suppression involves creating multiple TAG mutants, their screening and labeling optimization would be too time-consuming in primary neurons. Instead, this can be done in ND7/23 cells, a neuron-like cell line that is transfected with high efficiency. Compared to conventional host cell lines for amber codon suppression, such as HEK293 and COS7, the advantage of ND7/23 cells is that they have neuronal properties[63], hence they are frequently used as a model for differentiated neurons, for example, in electrophysiological studies. Thus, depending on the experimental design, ND7/23 cells can be used either as intermediate hosts for click labeling optimization (as applied in this manuscript) or as a main host for studies involving click-labeled neuronal proteins. Furthermore, we provide useful guidelines for the establishment and optimization of the method for labeling in primary neurons. In this regard, we compared different UAAs, different tetrazine dyes, and different single-color and multi-color labeling protocols. This is important for future applications and the successful implementation of this technique.

Finally, we show that click labeling can be combined with CRISPR/Cas9 genome engineering for labeling endogenous neuronal proteins. All studies published to date involving SPIEDAC labeling relied on overexpression of the target protein. Whereas this is a standard aspect of many cell biology experiments (not only for SPIEDAC-based labeling, but with other types of genetically encoded probes), protein overexpression can cause artefacts and cytotoxicity. Combining site-specific labeling with endogenous protein tagging would open new opportunities for protein studies, as it would allow labeling proteins at their endogenous expression levels. To this aim, we took advantage of two recently described methods for endogenous protein tagging in neurons[44,46]. However, as we discussed earlier, this brought about a new challenge. As amber suppression is not 100% efficient, to avoid affecting the levels of expression of endogenous NFL, instead of a site-specific TAG knock-in, we introduced the TAG amber codon at the end of the *Nefl* gene, in the form of a short linker sequence. With this strategy, we successfully combined genome editing and SPIEDAC reaction to fluorescently label proteins expressed under endogenous promoters. It is important to note that TKIT genome editing approach, as opposed to ORANGE, allowed us to have an intact *Nefl* open reading frame (ORF). Since it does not rely on editing in coding

regions, TKIT could be used for site-specific incorporation of TAG codon in the ORF of the target protein and subsequent site-specific click labeling of endogenous proteins. However, such premature stop codons would almost certainly lead to the reduction of endogenous protein levels. This knock-down would be caused partially by the suboptimal amber codon suppression efficiency and partially by the nonsense-mediated decay of premature stop codon-containing mRNAs. For this reason, C-terminal addition of TAG-containing small linkers and tags, as described here, represents a more optimal strategy when combining amber codon suppression and genome editing. However, the major limitation of the current approach is relatively low transfection and amber codon suppression efficiency of the knock-in constructs. By using more efficient viral vectors, this can be further refined and brought beyond the proof-of-principle level.

Compared with other methods, SPIEDAC labeling has a number of advantages. Firstly, this is the only method that allows site-specific labeling of proteins with single-residue precision. Clickable UAAs are L-lysine derivatives, adding only a few more atoms to the target protein, thus lowering the risk of functional impact. In addition, SPIEDAC labeling brings the dye as close as possible to the target protein. This is especially relevant for super-resolution imaging[1–5]. Conventional labeling approaches with large antibody complexes can place the fluorophore 20–30 nm away from the target protein. This results in "linkage error", which introduces localization artefacts and affects localization precision. Although this is not relevant for conventional microscopy with a resolution above 250 nm, it is a problem for super-resolution microscopy, especially if the resolution limit (5–10 nm) reaches the size of conventional labeling tags. Compared to larger labeling tags, such as primary–secondary antibody complexes, UAA-based labeling also allows fluorophores to be placed at a higher density, which is crucial for achieving optimal resolution[64]. However, it is important to note that this is a theoretical assumption that would require experimental verification. One previous study directly compared super-resolution imaging of click-labeled and antibody-labeled NMDA receptors and showed that click labeling allowed imaging of significantly higher receptor density[53]. As we did not quantify the labeling density of click-labeled NFL, we cannot claim with certainty if the achieved labeling density is indeed higher than what could be achieved with antibodies. Previously, fluorescent protein fusions were used for live-cell imaging of single NFL polymers[31]. Considering that click labeling is also stoichiometric (one tetrazine-dye molecule binds the UAA at a 1:1 ratio), we would expect similar fluorophore densities for these two labeling approaches. However, while GFP-fusions are intrinsically fluorescent, proteins bearing clickable UAAs are labeled with fluorescent dyes after translation. This could result in lower labeling densities than what is theoretically expected. An additional advantage of our approach is that it relies on the rapid and bioorthogonal SPIEDAC reaction. This reaction is not only compatible with live-cell imaging but is also favorable because of its fluorogenic character. Owing to the photophysical properties of certain fluorophores, their tetrazine derivatives are quenched, and their fluorescence is restored/enhanced upon reaction with a click-reactive partner. This property reduces background signal and makes SPIEDAC labeling with tetrazine dyes very attractive for live-cell and super-resolution microscopy applications[65,66]. Moreover, as also shown in our manuscript, SPIEDAC labeling brings flexibility in selecting the tetrazine dye. As different labeling and imaging approaches require dyes with different properties—for example, cell-impermeable dyes can be used for labeling of extracellular proteins, while cell-permeable dyes are needed for live-cell imaging of intracellular targets, photostable dyes are needed for

STED imaging, photoswitchable dyes for single-molecule localization techniques—the most suitable tetrazine dye can be chosen, depending on the desired application. A useful resource in this regard is a recent study by Beliu et al. in which spectroscopic and quenching properties, as well as suitability for confocal and super-resolution imaging of a wide range of tetrazine dyes were studied[66]. For standardization reasons, instead of live-cell click labeling, the authors performed post-fixation labeling of actin with TCO-containing phalloidin, followed by click chemistry reaction with tetrazine dyes. This allowed identification of dyes that can be used in a wash-free manner as well as dyes that give nonspecific fluorescent background in fixed cells. Although 22 known and new dyes were tested for the post-fixation labeling of TCO-phalloidin, only a subset of already known dyes, such as SiR, HMSiR, and Cy5, was subsequently used for live-cell labeling of TCO*A-Lys-containing extracellular and intracellular proteins in the presence of genetic code expansion machinery. In this manuscript, we show the suitability of several cell-permeable and impermeable tetrazine dyes for click labeling of TCO*A-Lys-containing intracellular proteins. In general, live-cell labeling with cell-permeable dyes is superior to post-fixation labeling with cell-impermeable dyes which showed higher background. In either case, the signal to noise was high enough even for advanced imaging applications, including STED microscopy. Finally, the large variety of fluorophores allows dual-color and potentially multi-color labeling studies. This can be achieved by using different tetrazine dyes, as we described here, or with mutually orthogonal SPIEDAC reactions, as shown previously[25]. Either way, with the addition of UAAs or tetrazine dyes at precisely defined time points, different populations of target proteins can be labeled. With the complex architecture of neurons in mind, we believe that this type of pulse–chase labeling will grant us novel insights into the fates of neuronal proteins.

However, there are also limitations that need to be considered when applying SPIEDAC-based protein labeling. Firstly, amber codon suppression relies on complex genetic engineering machinery that needs to be introduced into the host cell. Depending on the host cell line, this can result in low efficiency of UAA incorporation, especially in cells that are not readily transfected, such as non-dividing neurons. However, as we show with this study, even with conventional transfection methods, we achieved successful incorporation of UAAs in primary neurons. This might differ for other target proteins, but can be further improved by using viral vectors or transgenic animals for the expression of the orthogonal RS/tRNA pairs. Another limitation is that our approach relies on amber stop codon suppression, which shows position and sequence context-dependent efficiency[67]. Therefore, different TAG positions need to be tested to find the best-expressing mutant. This is also important for functional reasons, as site-specific mutations can alter the function of a target protein. In addition, the sequence-dependant varying levels of amber codon suppression efficiency could result in the translation of protein fragments that could potentially affect cellular processes. This is relevant for any protein of interest and is not only limited to our study. In the case of NFL, truncated fragments could have a dominant-negative effect on neurofilament assembly and could affect NFL function. However, our transfected neurons have normal morphology, they express clickable constructs over the course of several days without any signs of cytotoxicity, the neurofilament network is assembled properly and our click labeling co-localizes with the anti-NFL signal. This is in line with other studies using recombinant neurofilament constructs[31]. Furthermore, as we have shown by analyzing different TAG mutants and using more efficient approaches for amber codon suppression, such as mutant eRF1$^{E55D}$ and codon-optimized plasmids, the amount of

truncated proteins can be reduced to minute amounts (around 2% of total expressed protein), diminishing the possibility of fragment-induced disruptions. In addition, the site-specificity of the method offers a sufficient chance of finding the most suitable position which will keep the functional impact on the host cell to the minimum which is crucial for the successful application of SPIEDAC labeling for physiological studies. Finally, compared to other labeling tags, exchanging one of the residues with a UAA represents a much smaller modification of the protein of interest. However, free UAA molecules and components of orthogonal translational machinery could also get click labeled and thus contribute to the measured fluorescence, reducing the signal-to-noise ratio. This is particularly a problem for intracellular click labeling. As others and we have shown, the amount of background depends on the properties of UAAs and dyes. Different factors, such as UAA stability, reactivity towards tetrazines, hydrophilicity, need to be considered when choosing the UAA for optimal click labeling. We have previously shown the superiority of a mixture of isomers of TCO*-Lys over BCN-Lys for the labeling and microscopy studies of extracellular proteins[32], even though click product of TCO*-Lys with tetrazines (especially, substituted forms, such as methyl-tetrazines) could undergo elimination reaction. In this manuscript, we used a more stable isoform, TCO*A-Lys[68]. The literature on UAA stability is somewhat conflicting, mainly because different methods were used to study reactions rates and stability. Recent in cellulo experiments represent a very useful resource in this regard and "reidentify" previously neglected TCO4en/eq-Lys as a particularly stable UAA[41]. Furthermore, to reduce the background originating from free UAA molecules, UAAs that show a faster washout, such as BCN-Lys[39,40] and more hydrophilic dioxo-TCO-Lys[42] could be used. In any case, to incorporate these UAAs efficiently in neurons would require further optimizations including using recently developed PylRS synthetase for TCO4en/eq-Lys[41], selecting a more efficient synthetase for hydrophilic dioxo-TCO[42], as well as titrating the concentration of BCN-Lys to allow prolonged experiments with primary neurons. With regards to the tetrazine dyes, the number of cell-permeable dyes that can be used for intracellular labeling in living cells is still limited. However, thanks to the modality of click reaction, this labeling approach can easily take advantage of new discoveries in fluorescent dye synthesis, especially when it comes to new additions to the pool of membrane-permeable dyes for advanced imaging studies[69–72]. Furthermore, click labeling could be easily adapted to attach other types of probes (e.g., gold particles, affinity probes, PET, MRI imaging tracers) to the proteins of interest. Another potential limitation is that amber codon suppression might impact the translation of other proteins from genes that naturally use the amber codon as a stop codon. To what extent this happens in mammalian cells is unknown. However, as most of C-terminally extended proteins get eliminated from cells[45] and termination of translation depends on additional factors that even compete with the incorporation of UAAs at our desired site-specifically introduced TAG site, this type of background should be negligible and was not apparent from previous studies. As we also show in our manuscript, when working with transfected/overexpressed proteins, this type of background is not hindering advanced imaging. However, as levels of endogenous proteins are lower compared to transfected proteins, this background became more obvious when we combined it with genome engineering. This is a limitation that cannot be fully addressed with the current amber codon suppression approach, but other developments in the field offer promising alternatives. Although still at the proof-of-principle stage or with limited applicability in eukaryotic cells, alternative strategies using orthogonal organelles[73] for amber codon suppression, quadruplet codons[74,75] instead of stop codons, and engineered ribosomes[76], offer solutions to this problem.

In summary, we have established a minimally invasive approach for protein labeling in living primary neurons by combining two state-of-the-art technologies: incorporation of UAAs via genetic code expansion and ultrafast bioorthogonal SPIEDAC reactions. Site-specific labeling with UAA-based minimal tags expands the toolbox of available live-cell protein labeling methods in neurobiology. Furthermore, by establishing SPIEDAC labeling in complex, primary cells, such as neurons, we further expanded the portfolio of genetic code expansion-based applications in the field of cell biology. We believe that by complementing the currently available methods, intracellular and extracellular labeling by click chemistry will open new possibilities for advanced studies of neuronal cells involving neuronal protein labeling, trafficking in living neurons, labeling of endogenous proteins, as well as fixed-cell and live-cell super-resolution imaging.

## Methods

**Cell culture**. Mouse neuroblastoma × rat neuron hybrid ND7/23 cells were purchased from Sigma-Aldrich (ECACC 92090903). They were grown in high-glucose Dulbecco's Modified Eagle Medium (DMEM; ThermoFisher Scientific, cat. no. 41965062) supplemented with 10% heat-inactivated fetal bovine serum (FBS; ThermoFisher Scientific, cat. no. 10270106), 1% penicillin–streptomycin (PS; Sigma-Aldrich, cat. no. P0781), 1% sodium pyruvate (ThermoFisher Scientific, cat. no. 11360039) and 1% L-glutamine (ThermoFisher Scientific, cat. no. 25030024). FBS was inactivated by incubation at 56 °C for 30 min. Cells were passaged three times per week, and used for transfections at passages 3–15.

For microscopy experiments, ND7/23 cells were seeded on eight-well Lab-Tek II chambered cover glasses (German #1.5 borosilicate glass; ThermoFisher Scientific, cat. no. 155409) at a density of 25,000 cells per well. Prior to cell seeding, the cover glasses were coated with a 10 μg/ml solution of poly-D-lysine (Sigma-Aldrich, cat. no. P6407) in double-distilled water (ddH₂O) for a minimum of 4 h at room temperature (RT). Chambered cover glasses were washed three times with ddH₂O and allowed to dry prior to cell seeding. For lysis and western blot analysis, ND7/23 cells were seeded in 6-well plates (Greiner Bio-One, cat. no. 657160), at a density of 200,000 cells per well.

Primary mouse cortical neurons (MCNs) from C57BL/6 embryonic day 17 were purchased from ThermoFisher Scientific (cat. no. A15586). They were thawed and cultured according to the manufacturer's recommendation in a B-27 Plus Neuronal Culture System consisting of Neurobasal Plus (NB Plus) medium and B27 Plus supplement (ThermoFisher Scientific, cat. no. A3653401). Culturing medium was prepared by adding 2% of B27 Plus supplement and 1% of PS to Neurobasal Plus + (NB Plus +). For widefield and confocal microscopy experiments, MCNs were seeded on eight-well Lab-Tek II chambered cover glasses at a density of 90,000–110,000 cells per well. For experiments that did not require transfection, MCNs were seeded at a density of 70,000 cells per well. For STED imaging, MCNs were seeded on eight-well μ-slides with glass bottoms (Ibidi cat. no. 80827), at a density of 100,000 cells per well. For experiments that involved isolation of neuronal genomic DNA or total RNA, MCNs were seeded in 12-well plates (Corning Incorporated, cat. no. 3512), at a density of 500,000–1 million cells per well. The bottoms of the Lab-Tek chambers, μ-slides, and 12-well plates were pre-coated with a 20 μg/ml solution of poly-D-lysine in ddH₂O for 2 h at RT. Prior to cell seeding, they were washed three times with ddH₂O, allowed to dry, and then pre-incubated for at least 30 min with NB Plus + medium. During the culturing of the MCNs, half the NB Plus + medium was exchanged twice per week.

**Constructs, cloning, and mutagenesis**. The cDNA encoding for mouse neurofilament light chain (NFL) was amplified from the vector pmNFL (a gift from Anthony Brown, Addgene plasmid #83127; http://n2t.net/addgene:83127; RRID: Addgene_83127)[77] and initially cloned in an mEGFP-N1 plasmid (a gift from Michael Davidson, Addgene plasmid #54767; http://n2t.net/addgene:54767; RRID: Addgene_54767) using HindIII (ThermoFisher Scientific, cat. no. FD0504) and ApaI (ThermoFisher Scientific, cat. no. FD1414) enzymes. In the resulting construct, the TAG amber stop codon was introduced at positions K211, K363, R438, and K468 of the NFL cDNA, via PCR-based site-directed mutagenesis. After the mutagenesis, GFP was excised from all constructs using the enzymes BamHI (ThermoFisher Scientific, cat. no. FD0054) and NotI (ThermoFisher Scientific, cat. no. FD0595), and replaced by a double-stranded DNA oligonucleotide containing the FLAG tag sequence (DYKDDDDK). The FLAG tag oligonucleotide was synthesized by Sigma-Aldrich as two complementary single-stranded oligonucleotides (Supplementary Table 3).

FLAG-NFL$^{TAG}$ constructs were cloned by excising the NFL$^{WT}$-FLAG-encoding sequence from the pCMV backbone and replacing it with the FLAG-NFL$^{TAG}$-encoding sequence, using the enzymes HindIII and NotI. The DNA encoding

FLAG-NFL[TAG] was amplified by PCR from the NFL[TAG]-GFP plasmids using the forward primer that contained the FLAG-encoding sequence (Supplementary Table 3). FLAG-NFL[WT] construct was cloned next by using enzymes KpnI (ThermoFisher Scientific, cat. no. FD0524) and PvuI (ThermoFisher Scientific, cat. no. FD0624) to excise the K363TAG-containing DNA fragment from the FLAG-NFL[K363TAG] construct and replace it with the corresponding WT sequence.

Together with NFL, we co-transfected neurofilament medium chain (NFM) cDNA-containing plasmid pmNFM (a gift from Anthony Brown, Addgene plasmid #83126; http://n2t.net/addgene:83126; RRID: Addgene_83126)[77].

For the experiments involving amber codon suppression of overexpressed NFL[TAG] mutants, we used a previously published pcDNA3.1/Zeo(+) plasmid[26] containing a sequence that encodes *Methanosarcina mazei* pyrrolysyl-tRNA synthetase with a nuclear export signaling sequence and Y306A, Y384F substitutions (NES PylRS[AF]), and one copy of tRNA[Pyl][CUA] under the control of the U6 promoter (a kind gift from Edward Lemke's laboratory, EMBL, Heidelberg, and IMB, Mainz).

For the experiments involving amber codon suppression of endogenous NFL and βIII-tubulin, as well as western blot analysis, we used a pcDNA3.1/Zeo(+) plasmid containing codon-optimized sequence that encodes *Methanosarcina mazei* NES PylRS[AF] and one copy of tRNA[Pyl][CUA]. The codon-optimized sequence encoding the NES PylRS[AF] was synthesized by GenScript and cloned into the pcDNA3.1/Zeo(+) vector. We subsequently added tRNA[Pyl][CUA] and the U6 promoter or the 4xU6-M15tRNA[CUA][27] cassette upstream of the CMV promoter in the reverse direction by cloning, using BglII (ThermoFisher Scientific, cat. no. FD0083) and MfeI (ThermoFisher Scientific, cat. no. FD0753) enzymes. The U6 promoter-tRNA cassette synthesized by GenScript or pNEU-hMbPylRS-4xU6M15 plasmid (a gift from Irene Coin, Addgene plasmid #105830; http://n2t.net/addgene:105830; RRID:Addgene_105830)[27] were used as a template for the cloning. For the amber codon suppression of endogenous proteins as well as western blot analysis, we also co-transfected cells with eukaryotic release factor 1 E55D mutant (eRF1[E55D]). This plasmid was cloned by Christopher D. Reinkemeier in Edward Lemke's laboratory.

For the labeling of endogenous βIII-tubulin, we used a pORANGE Tubb3-GFP KI plasmid (a gift from Harold MacGillavry, Addgene plasmid #131497; http://n2t.net/addgene:131497; RRID: Addgene_131497)[44]. For the optimization of genetic code expansion of endogenous βIII-tubulin, we replaced GFP from this construct with GFP[Y39TAG] by cloning with HindIII (ThermoFisher Scientific, cat. no. FD0504) and XhoI (ThermoFisher Scientific, cat. no. FD0694) restriction sites.

In order to label endogenous NFL, we designed and cloned target and donor sequences following the previously published protocol[44]. The NFL target sequence GAGTGCTGGAGAGGAGCAGG (https://wge.stemcell.sanger.ac.uk//crispr/377510968) was selected using the Ensembl browser[78] [Ensembl release 102, November 2020; *Mus musculus* version 102.38 (GRCm38.p6), Chromosome 14: 68,087,408–68,087,430] based on available PAM sites at the end of the *Nefl* gene. The integration site is Q537 and the knock-in results in the deletion of six C-terminal amino acids of the NFL protein. The target sequence was subsequently cloned into the pORANGE cloning template vector (a gift from Harold MacGillavry Addgene plasmid #131471; http://n2t.net/addgene:131471; RRID: Addgene_131471)[44] using the BbsI enzyme (ThermoFisher Scientific, cat. no. FD1014) and single-stranded DNA oligonucleotides synthesized by Sigma-Aldrich. In the next step, we cloned a donor sequence containing *linker*-3xFLAG (GSAGSA-DYKDHDGDYKDHDIDYKDDDDK) or *linker[A6TAG]*-3xFLAG (GSAGS*-DYKDHDGDYKDHDIDYKDDDDK) into the resulting plasmid (pORANGE NFL KI), using the enzymes HindIII and BamHI. The donor sequences were amplified by PCR from existing plasmids.

For labeling of the endogenous NFL via Targeted Knock-In with Two guides approach (TKIT)[46] we selected two target sequences in the *Nefl* gene, using the Ensembl browser[78]. First target sequence (gRNA1; https://wge.stemcell.sanger.ac.uk//crispr/377510942) is located in the last intron (intron 3) of *Nefl* gene, 114 bp upstream of the start of the last exon (exon 4). Second target sequence (gRNA2; https://wge.stemcell.sanger.ac.uk//crispr/377510985) is located in the 3′ untranslated region (UTR) of the *Nefl* gene, 107 bp downstream of the STOP codon. Both target sequences were synthetized by Sigma-Aldrich as single-stranded DNA oligonucleotides and were separately cloned into the pORANGE cloning template vector using the BbsI enzyme. The plasmid containing gRNA2 was subsequently used as a template for PCR amplification of the U6 promoter-gRNA2 cassette, which was then cloned in the multiple cloning site of the plasmid containing gRNA1, using enzymes XbaI (ThermoFisher Scientific, cat. no. FD0684) and SalI (ThermoFisher Scientific, cat. no. FD0644). The resulting construct contains U6 promoter-gRNA1 followed by U6 promoter-gRNA2 as well as spCas9 expressed from the CAG promoter. Donor sequences for TKIT knock-in were synthetized by Eurofins Genomics and contain the following sequences: reverse complement sequence of gRNA2 followed by a part of *Nefl* intron 3, whole *Nefl* exon 4 with the addition of linker[WT]-3xFLAG sequence or linker[A6TAG]-3xFLAG sequence, a part of the 3′ UTR and a reverse complement sequence of gRNA1. Subsequently, both WT- and A6TAG-linker-containing donor sequences were cloned upstream of the CMV promoter in the pcDNA3.1/Zeo(+) and in the pcDNA3.1/Zeo(+)-mCherry vectors. For the experiments involving the combination of TKIT-based knock-in and amber codon suppression, pcDNA3.1/Zeo(+) vector containing tRNA[Pyl][CUA] and codon-optimized sequence encoding NES PylRS[AF] was modified to include internal ribosomal entry site (IRES) followed by eRF1[E55D]-encoding sequence.

All primer and oligonucleotide sequences used for cloning and mutagenesis are listed in Supplementary Table 3. Donor sequences for ORANGE- and TKIT-based knock-in and primers used for PCR amplification and sequencing of genomic DNA and cDNA are listed in Supplementary Table 4.

**UAAs, tetrazine derivatives of fluorescent dyes, and antibodies**. In this study, the following unnatural amino acids (UAAs) were used: *trans*-cyclooct-2-en-L-lysine (TCO*A-Lys; Sirius Fine Chemicals, SICHEM, cat. no. SC-8008), *trans*-cyclooct-4-en-L-lysine (TCO4en/eq-Lys; a kind gift from Edward Lemke's laboratory, also available from SICHEM, cat. no. SC-8060) and *endo*-bicyclo[6.1.0]non-yne-lysine (endo-BCN-Lys; SICHEM cat. no. SC-8014). For SPIEDAC labeling, the following tetrazine derivatives of fluorescent dyes were used: ATTO655-methyltetrazine (ATTO655-me-tz; ATTO-TEC GmbH, cat. no. AD 655-2502), ATTO488-tetrazine (ATTO488-tz; Jena Bioscience, cat. no. CLK-010-02), CF500-methyltetrazine (CF500-me-tz; Biotium cat. no. 96029), CF650-methyltetrazine (CF650-me-tz; Biotium cat. no. 96036), Janelia Fluor 646-methyltetrazine (JF646-me-tz; Jena Bioscience custom synthesis), Janelia Fluor 549-tetrazine (JF549-tz; Tocris cat. no. 6502), silicon rhodamine-tetrazine (SiR-tz; SpiroChrome cat. no. SC008), Alexa Fluor 647-tetrazine (AF647-tz; a kind gift from Edward Lemke's laboratory), TAMRA-tetrazine (TAMRA-tz; Jena Bioscience cat. no. CLK-017-05), and BODIPY-tetrazine (BODIPY-tz; Jena Bioscience cat. no. CLK-036-05). For immunocytochemistry and western blot, the following antibodies were used: rabbit anti-FLAG antibody (Merck Millipore cat. no. F7425), mouse anti-FLAG M2 antibody (Sigma-Aldrich, cat.no. F1804), mouse anti-neurofilament 70 kDa antibody, clone DA2 (Merck Millipore cat. no. MAB1615), mouse anti-βIII-tubulin antibody (BioLegend, cat. no. 801202), goat anti-rabbit Alexa Fluor (AF) 488 Plus (ThermoFisher Scientific, cat. no. A32731), goat anti-rabbit AF555 (ThermoFisher Scientific, cat. no. A21429), goat anti-mouse AF555 (ThermoFisher Scientific, cat. no. A21424), goat anti-rabbit AF647 Plus (ThermoFisher Scientific, cat. no. A32733), goat anti-mouse AF488 Plus (ThermoFisher Scientific, cat. no. A32723), goat anti-mouse AF647 Plus (ThermoFisher Scientific, cat. no. A32728), goat anti-rabbit horseradish peroxidase (HRP; ThermoFisher Scientific, cat. no. A16104), and goat anti-mouse HRP (ThermoFisher Scientific, cat. no. A16072).

**Transfections**. Both ND7/23 cells and MCNs were transfected using the Lipofectamine 2000 transfection reagent (ThermoFisher Scientific, cat. no. 11668027). ND7/23 cells were transfected 14–20 h after seeding into an eight-well Lab-Tek chambered slide, with a slightly modified manufacturer's protocol using a DNA/Lipofectamine 2000 ratio of 1 μg:2.4 μl and up to 0.625 μg of total DNA per well. Immediately after transfection, a stock solution of UAA (100 mM in 0.2 M NaOH containing 15% DMSO) was diluted 1:4 in 1 M HEPES (ThermoFisher Scientific, cat. no. 15630080) and added to cells to a final concentration of 250 μM for TCO*A-Lys and TCO4en/eq-Lys, and 1 mM for endo-BCN-Lys. The medium was replaced after incubation for 6 h (37 °C, 5% CO₂), and the HEPES-diluted UAA was again added before cells were incubated overnight (37 °C, 5% CO₂).

For the transfection of MCNs, we adapted a previously published protocol[79] using Lipofectamine 2000. As was done for ND7/23 cells, we used a DNA/Lipofectamine 2000 ratio of 1 μg:2.4 μl. The total amount of DNA per well in an eight-well Lab-Tek chambered slide was up to 1.25 μg. DNA and Lipofectamine 2000 solutions were prepared in NB Plus medium with 1% of PS. Prior to their addition to the cells, transfection solutions were mixed with an equal volume of warm NB Plus medium containing 4% B27 Plus to obtain a final B27 Plus content of 2%. These final transfection mixtures were then warmed by incubation for 5 min (37 °C, 5% CO₂). The culturing medium was aspirated from neurons and retained for use as conditioned medium (CM), and warm transfection mixture was added dropwise to cells. After incubation for 4–6 h, the transfection medium was aspirated, and the retained CM was added back to cells. If the transfection was performed on the day of medium change, half the volume of CM was put back to the cells and topped up with fresh NB Plus + medium. Afterward, 100 mM TCO*A-Lys, endo-BCN-Lys or TCO4en/eq-Lys stock (in 0.2 M NaOH containing 15% DMSO) was diluted 1:4 in 1 M HEPES and added to neurons, to a final concentration of 250 μM. MCNs were incubated (37 °C, 5% CO₂) for a minimum of two days prior to click labeling. The exact labeling time points are described in figure legends. For NFL[TAG] overexpression experiments, MCNs were transfected at day in vitro (DIV) 8, and for the labeling of endogenous NFL with pORANGE or TKIT vectors, MCNs were transfected at DIV5. For the labeling of endogenous βIII-tubulin with pORANGE vectors, MCNs were transfected at DIV3.

For lysis and western blot analysis, ND7/23 cells were transfected 14–20 h after seeding in 6-well plates using the Lipofectamine 2000 transfection reagent. DNA/Lipofectamine 2000 ratio was 1 μg:2.4 μl and the total DNA amount was 5.25 μg/well. Alternatively, cells were transfected using the calcium phosphate method. Briefly, per well of a 6-well plate, 25 μl of 1 M calcium chloride (Sigma-Aldrich, cat. no. C5670) was mixed with 5.25 μg of DNA, and sterile water was added up to 100 μl. This mixture was added dropwise to 100 μl of 2× HBS (50 mM HEPES, 280 mM NaCl, 1.5 mM Na₂HPO₄, pH 7), slowly, while flicking the tube to mix the solutions. After 10 min of incubation, transfection mixtures were added dropwise to cells. Immediately after transfection, a stock solution of TCO*A-Lys (100 mM in 0.2 M NaOH containing 15% DMSO) was diluted 1:4 in 1 M HEPES and added to cells to a final concentration of 250 μM. The medium was replaced after incubation

for 6 h (37 °C, 5% CO$_2$), and the HEPES-diluted TCO*A-Lys was again added before cells were incubated overnight (37 °C, 5% CO$_2$).

**Single-color click chemistry labeling in live ND7/23 cells and neurons with cell-permeable dyes.** ND7/23 cells expressing NFL$^{WT}$-FLAG or NFL$^{TAG}$-FLAG mutants were labeled by SPIEDAC click chemistry after overnight (18-20 h) incubation with the UAA. To wash out excess UAA, medium containing the UAA was removed, cells were washed twice with culturing medium and then incubated 2 h in a fresh culturing medium. Cells were washed once more with culturing medium and incubated for 10 min (37 °C, 5% CO$_2$) with the tetrazine dye diluted in culturing medium. The concentration of tetrazine dyes was 5 μM, except for JF549-tz, which was used at a concentration of 2.5 μM. After incubation for 10 min, the dye-containing medium was aspirated, cells were washed twice and then incubated for 2 h in fresh culturing medium. Afterward, the culturing medium was aspirated, cells were washed once with 0.01 M phosphate-buffered saline (PBS; 137 mM NaCl, 10 mM Na$_2$HPO$_4$, 1.8 mM KH$_2$PO$_4$, 2.7 mM KCl, pH 7.4) and fixed with 4% paraformaldehyde (PFA; Sigma-Aldrich, cat. no. 158127) in 0.1 M phosphate buffer (PB) for 15 min at RT. After fixation, the FLAG tag was labeled by immunocytochemistry.

Live-cell click chemistry labeling of MCNs was performed 2–3 days (overexpression experiments) or 6 days (labeling of endogenous NFL with pORANGE or TKIT vectors) after transfection. First, medium containing the UAA was removed, neurons were washed twice with fresh NB Plus +, and then incubated for 2–3 h (37 °C, 5 % CO$_2$) in a 1:1 mixture of fresh NB Plus + and CM (collected either on the day of transfection, or from neurons cultured only for this purpose). Afterward, neurons were washed once more with fresh NB Plus + and incubated with the tetrazine dye diluted in fresh NB Plus + for 10 min (37 °C, 5% CO$_2$). The concentrations of dyes were the same as for the labeling in ND7/23 cells. After the labeling period, neurons were washed twice with fresh NB Plus + and incubated for 2–3 h in a 1:1 mixture of fresh NB Plus + and CM. For the click chemistry labeling background measurements, neurons were incubated either 2 h or 10 h after the labeling. The culturing medium was then aspirated, and neurons were fixed with 4% electron microscopy grade PFA (Electron Microscopy Sciences, cat. no. 15710) diluted in PEM buffer (80 mM PIPES, 2 mM MgCl$_2$, 5 mM EGTA, pH 6.8). After fixation, depending on the experiment, and as described in the corresponding figure legends, immunocytochemistry was performed.

**Single-color click chemistry labeling in ND7/23 cells and neurons after fixation.** Labeling with the cell-impermeable tetrazine dyes ATTO655-me-tz, ATTO488-tz, and AF647-tz was performed after cell fixation. First, the UAA was removed and cells were washed according to the washing procedure used for live-cell labeling. After 2–4 h of washing, the medium was aspirated, and ND7/23 cells were rinsed with PBS and fixed with 4% PFA in 0.1 M PB, whereas neurons were fixed without PBS rinsing with 4% PFA in PEM buffer for 15 min at RT. Cells were permeabilized with 0.1% Triton X-100 (Sigma-Aldrich, cat. no. X-100) in PBS for 10 min at RT. Tetrazine dyes were diluted to a working concentration of 0.5–2.5 μM in PBS. Cells were rinsed with PBS and labeled with dyes for 10 min at 37 °C. After labeling, cells were rinsed three times with PBS and incubated on a shaker at RT for 20–30 min. Afterward, the FLAG tag was labeled by immunocytochemistry.

**Pulse–chase click chemistry labeling of two NFL populations in neurons.** Neurons were labeled with the first tetrazine dye (BODIPY-tz) two days after transfection, following the same protocol as for the single-color live click chemistry labeling. After labeling with the first dye, neurons were washed for 2–3 h in a 1:1 mixture of fresh NB Plus + and CM, and then incubated with TCO*A-Lys for a further 3 h or 2 days. After that time, neurons were labeled with the second dye (SiR-tz) following the same protocol as above. Alternatively, after labeling with the first dye, neurons were washed in a 1:1 mixture of fresh NB Plus + and CM for 1 day, and then incubated with TCO*A-Lys for 1 day. Afterward, neurons were washed, labeled with SiR-tz, and fixed immediately after labeling. For the controls of two NFL population labeling, after labeling with the first dye (BODIPY-tz), neurons were either labeled immediately with the second dye (SiR-tz), or incubated without TCO*A-Lys for 2 days, and then labeled with the second dye (SiR-tz). After labeling with the second dye, neurons were either fixed and stained using anti-FLAG immunocytochemistry or imaged live with confocal scanning microscopy.

For STED imaging experiments involving labeling of two NFL populations and oxidative injury, we established a slightly different protocol. Two days after transfection, neurons were labeled with SiR-tz by following the same protocol as for the single-color live click chemistry labeling. Then, neurons were washed for 3 h and incubated for 30 min (37 °C, 5% CO$_2$) with either 25 μM spermine-NONOate (a nitric oxide donor; Cayman Chemical, cat. no. 82150) or 25 μM sulpho-NONOate (control compound; Cayman Chemical, cat. no. 83300), in the presence of TCO*A-Lys. After injury, neurons were rinsed with warm NB Plus + and incubated for 2 days with TCO*A-Lys. Then, neurons were fixed and labeled with 1-1.5 μM ATTO488-tz following the fixed-cell labeling protocol described above.

**MitoTracker and LysoTracker labeling.** For the experiments with MitoTracker and LysoTracker labeling, neurons were transfected and labeled with SiR-tz as described above. After washing for 2–3 h, 250 μl of NB Plus + medium containing 100 nM MitoTracker Orange (ThermoFisher Scientific, cat. no. M7510) and 400 nM LysoTracker Green (Cell Signaling Technology, cat. no. 8783 S) was added to the wells that already contained 250 μl of medium, for final concentrations of 50 and 200 nM for MitoTracker and LysoTracker, respectively. For the additional control experiments (data shown in Supplementary Fig. 9) involving tetrazine-dye and LysoTracker labeling, MCNs were seeded at a density of 70,000 cells per well and were not transfected. At DIV8, HEPES-diluted TCO*A-Lys was added to cells at a concentration of 250 μM. After 3 days of incubation with TCO*A-Lys, MCNs were labeled with either BODIPY-tz or SiR-tz, washed for 3 h, and then labeled as described above with 200 nM of either LysoTracker Green or LysoTracker Deep Red (ThermoFisher Scientific, cat. no. L12492). Neurons were incubated with MitoTracker and LysoTracker dyes for 30 min and rinsed twice with NB Plus +. Immediately afterward, NB Plus + was replaced by Hibernate E medium (Brain Bits LLC, cat. no. HELF) containing 1% PS and 2% B27 Plus, and neurons were imaged live with confocal scanning microscopy.

**Immunocytochemistry staining.** For anti-NFL and anti-FLAG immunocytochemical staining, cells and neurons were fixed as described above, then washed three times (5 min each wash) with PBS. Afterward, cells were incubated for 1 h at RT with a blocking serum containing 3% bovine serum albumin (BSA; Sigma-Aldrich, cat. no. A9647), 10% goat serum (GS; ThermoFisher Scientific, cat. no. 16210072), and 0.2% Triton X-100 diluted in PBS. For ORANGE and TKIT knock-in efficiency experiments, cells were first permeabilized with 0.1% Triton X-100 for 10 min at RT, briefly washed, and blocked with 3% BSA, 10% GS solution in PBS for 1 h at RT. Primary and secondary antibodies were diluted in the corresponding blocking serum (with or without 0.2% Triton X-100). Rabbit anti-FLAG antibody was used at a dilution of either 1:1000 (overexpression experiments) or 1:2000 (endogenous NFL labeling experiments with pORANGE/TKIT vectors), while mouse anti-FLAG M2 antibody was used at a dilution of 1:2000. Mouse anti-NFL antibody and all secondary antibodies were used at a dilution of 1:500. Cells were incubated with primary antibodies either overnight at 4 °C or for 1 h at RT, then washed three times (5 min each wash) with PBS and incubated for 1 h at RT with the secondary antibodies. Afterward, cells were washed three times (5 min each wash) with PBS and either imaged immediately, or stored at 4 °C until imaging.

**Cell lysis and western blot analysis.** One day after the transfection, ND7/23 cells were collected from 6-well plates and lysed using RIPA buffer (12.5 mM Tris hydrochloride, 37 mM NaCl, 3 mM sodium deoxycholate, pH 8) containing 1:50 protease inhibitor cocktail (Sigma-Aldrich, cat. no. P8340), 1 mM phenylmethanesulfonyl fluoride (Sigma-Aldrich, cat. no. P7626), and 50 mM sodium fluoride (Sigma-Aldrich, cat. no. S7920). Alternatively, cells were lysed with RIPA buffer containing 1:100 Halt™ Protease and Phosphatase Inhibitor Cocktail (Thermo-Fisher Scientific, cat. no. 78440) and 1 mM phenylmethanesulfonyl fluoride. Cells were incubated with the lysis buffer for 40 min on ice, then centrifuged at 4 °C for 30 min at 18,000×g. Protein-containing supernatants were transferred into clean tubes and protein concentration was measured using the Bradford reagent (Sigma-Aldrich, cat. no. B6916).

For SDS-PAGE electrophoresis, samples were mixed with Laemmli buffer (BioRad, cat. no. 1610747) and denatured for 5 min at 95 °C. Samples were loaded on NuPAGE™ 4-12% Bis-Tris Protein Gels (ThermoFisher Scientific, NP0329), 20 μg of protein/well. Electrophoresis was performed in 1× MOPS buffer (ThermoFisher Scientific, cat. no. NP0001) for 1 h at 150 V. After electrophoresis, proteins were transferred to a 0.2 μm nitrocellulose membranes (BioRad, cat. no. 1704158) by semi-dry Trans Blot Turbo transfer (BioRad, cat. no. 1704150), 7 min at 25 V and 2.5 A. Membranes were then stained with Ponceau S solution, imaged, and washed in water. Afterward, membranes were blocked for 1 h at RT in 5% milk (w/v) in TBS buffer (Tris-buffered saline; 20 mM Tris, 150 mM NaCl, pH 7.6) containing 0.05% Tween 20 (TBS-T; Sigma-Aldrich, cat. no. P7949). After blocking, membranes were incubated with primary antibodies diluted 1:5000 in 3% BSA in TBS-T for 1 h at RT on a rotating shaker. Primary antibodies were washed three times in TBS-T, 5 min each wash. Membranes were incubated with HRP-conjugated secondary antibodies, diluted 1:5000 in 3% BSA in TBS-T for 1 h at RT on a rotating shaker. Secondary antibodies were washed two times in TBS-T, each wash 5 min, and once with TBS before addition of Clarity Western ECL substrate (BioRad, cat. no. 1705060). Chemiluminescence was visualized using Azure 600 imager (Azure Biosystems). For the samples containing NFL-FLAG, membranes were first stained with anti-FLAG antibody, imaged, subsequently stained with anti-βIII-tubulin antibody, and imaged again. Before anti-βIII-tubulin labeling of FLAG-NFL-containing samples, membranes were stripped by incubating two times for 10 min in the stripping buffer (200 mM glycine, 3.5 mM sodium dodecyl sulfate, 1% Tween 20, pH 2.2). Membranes were then washed two times with TBS, each wash 5 minutes, and two times with TBS-T, each wash 10 min. After the stripping, membranes were blocked and labeled with primary (1:5000 dilution) and secondary antibodies (1:5000 dilution) as described above.

Western blot analysis was done using AzureSpot software (Azure Biosystems). Total volume of anti-FLAG bands was measured automatically and normalized to the volume of corresponding tubulin βIII bands, which served as a loading control.

The percentages of full-length and truncated FLAG-NFL[TAG] were calculated automatically as band percentage in AzureSpot. Data were collected from three independent experiments, and are shown as average percentages, with the corresponding SEM values in Supplementary Fig. 2. Supplementary Tables 1 and 2 contain the full data sets that were used for the analysis. Uncropped scans of the blots are provided at the end of the Supplementary Information. Raw images are provided on Figshare under following https://doi.org/10.6084/m9.figshare.c.5749409[80].

**Quantification of CRISPR/Cas9 knock-in efficiency**. For the quantification of ORANGE-mediated knock-in efficiency, MCNs were transfected at DIV5 with a 1:1 ratio of pORANGE NFL linker[WT]-3xFLAG KI and pcDNA3.1/Zeo(+)-mCherry constructs. MCNs were fixed with 4% PFA in PEM buffer 24 h, 72 h, and 144 h after transfection, stained with mouse anti-FLAG antibody, and imaged with widefield microscopy. Transfected cells were identified based on their mCherry signal and counted as knock-in positive or negative based on the presence or absence of the FLAG signal. 534 mCherry[+] cells per time point were collected from three experiments.

For the quantification of TKIT-mediated knock-in efficiency, MCNs were transfected at DIV5 with a 1:1 ratio of a plasmid containing gRNA1, gRNA2, and spCas9, and with pcDNA3.1/Zeo(+)-mCherry plasmid containing the linker[WT]-3xFLAG donor sequence. After 144 h, MCNs were fixed, immunostained, and imaged in the same way as for the ORANGE knock-in efficiency experiments. 600 mCherry[+] cells were collected from two experiments.

**Assessment of ORANGE knock-in specificity**. For the assessment of ORANGE-mediated knock-in specificity, MCNs were transfected at DIV5 with pORANGE NFL linker[WT]-3xFLAG KI plasmid and incubated for 4 days. Genomic DNA was extracted using the PureLink Genomic DNA mini kit (ThermoFisher Scientific, cat. no. K182001) and used as a template for touchdown PCR with primers amplifying 5′ and 3′ junction of the integrated donor sequence. PCR products were separated on 1% agarose gel, extracted using PureLink Quick Gel Extraction Kit (Thermo-Fisher Scientific, cat. no. K210012), and sequenced (LGC Genomics GmbH, Germany). Correct insertion of the donor sequence was confirmed by sequence analysis, using the Vector NTI Advance software (Life Technologies).

Primers used for touchdown PCR and sequencing are listed in Supplementary Table 4.

**Analysis of *Nefl* mRNA after TKIT-mediated knock-in**. For the analysis of proper *Nefl* mRNA splicing after TKIT-mediated knock-in, MCNs were transfected at DIV5 with a 1:1 ratio of a plasmid containing gRNA1, gRNA2, and spCas9, and with pcDNA3.1/Zeo(+) plasmid containing the linker[WT]-3xFLAG donor sequence. After 144 h, total RNA was isolated from neurons using RNAqueous Micro Total RNA Isolation Kit (ThermoFisher Scientific, cat. no. AM1931). Total RNA was then used as a template for cDNA synthesis using the oligo-dT primer and a SuperScript™ IV First-Strand Synthesis System (ThermoFisher Scientific, cat. no. 18091150). Resulting cDNA was amplified by PCR, using the primers that flank the splice junction between *Nefl* exons 3 and 4, and primers that flank the 3′ junction of the integrated donor sequence. PCR products were then separated on 1% agarose gel, extracted using PureLink Quick Gel Extraction Kit (ThermoFisher Scientific, cat. no. K210012), and sequenced (LGC Genomics). Sequence analysis showed that the donor sequence is correctly integrated in the *Nefl* gene and that splicing between exons 3 and 4 is unaffected by the donor sequence integration.

Primers used for PCR and sequencing are listed in Supplementary Table 4.

**Widefield imaging**. Widefield epifluorescence imaging was performed on an inverted Nikon Eclipse Ti2-E microscope (Nikon Instruments), equipped with XY-motorized stage, Perfect Focus System, and an oil-immersion objective (Apo 60×, NA 1.4, oil). Setup was controlled by NIS-Elements AR software (Nikon Instruments). Fluorescent light was filtered through 488 (AHF; EX 482/18; DM R488; BA 525/45), 561 (AHF; EX 561/14; DM R561; BA 609/54), and Cy5 (AHF; EX 628/40; DM660; BA 692/40) filter cubes. A fluorescent lamp (Lumencor Sola SE II) was used as a light source and emitted light was imaged with ORCA-Flash 4.0 sCMOS camera (Hamamatsu Photonics). Images were acquired at 16-bit depth, 1024 × 1024 pixels, and pixel size 0.27 μm.

**Confocal imaging of fixed and live cells**. Confocal imaging was performed on an LSM 710 confocal scanning microscope (Zeiss, Oberkochen, Germany), controlled by ZEN 2011 (Zeiss) software, equipped with a Plan-Apochromat 63× objective (NA 1.4, oil), 488, 561 and 633 nm laser lines, and continuous spectral detection. Images were acquired at 16-bit depth, 1024 × 1024 pixels, pixel size 0.132 μm, with 2× line averaging and a pixel dwell time of 6.3 μs, either as a single plane or as a Z-stack with 0.37 μm step size. In all channels, pinhole was set to 1 Airy unit. Emission light was collected sequentially, according to the emission spectra of the fluorophores used.

For the live-cell imaging, a temperature module was used and cells were placed in a heating insert (PeCon, Erbach Germany), which had been equilibrated to 37 °C. The medium used for imaging was Hibernate E medium containing 1% PS and 2% B27 Plus.

**STED imaging**. Super-resolution STED imaging was performed on a Leica TCS SP8 microscope (Leica Microsystems IR GmbH, Germany), using an HC PL APO CS2 100×/1.4 oil objective, hybrid detectors, and the following laser lines: 488 and 635 nm pulsed excitation lasers, as well as continuous 592 nm and gated pulsed 775 nm depletion lasers. Setup is controlled by LAS X (Leica) software. Excitation laser power and detector gain were adjusted for each image individually by avoiding over-saturation, according to the high and low pixel values. Emission light was collected sequentially, according to the emission spectra of the fluorophores used, for example, 500–580 nm (for ATTO488, AF488) or 645–760 nm (for SiR, CF650). Depletion lasers were used at 35–45% of maximum power. Images were acquired at 8-bit depth, image size 2048 × 2048 pixels, and pixel size 13-14 nm. For the confocal imaging, line averaging was 3× and frame accumulation 3×, whereas for STED imaging frame averaging was from 2 to 4×, and line accumulation 4×.

**Image processing**. Raw images were processed in Fiji software[81]. Widefield images were processed by linear adjustment of brightness and contrast and saved as TIFF files. Confocal Z-stack images were converted into maximum intensity projections, the brightness and contrast were adjusted linearly, and images were saved as TIFF files.

Colocalization analysis of anti-FLAG, anti-NFL and click chemistry SiR-tz labeled NFL was performed using the EzColocalization plugin for Fiji[82]. Full 16-bit images were imported in Fiji and cropped to contain only the region of interest (ROI). Each ROI is outlined by a dashed box in the corresponding figures. Fluorescently labeled regions were identified with automatic default thresholding, and analyzed using the Costes' algorithm for the calculation of Pearson's correlation coefficient. Colocalization data were visualized as cell pixel intensity scatterplots, xy coordinates were exported to Microsoft Excel and used for the generation of scatterplots. Scatterplots were further imported in Adobe Illustrator for the final presentation in figures.

STED images and the corresponding confocal images were deconvolved using Huygens deconvolution software (SVI, Netherlands), using Classical Maximum Likelihood Estimation (CMLE) algorithm. The signal-to-noise ratio was set to 20 for confocal and to 7 for STED images, and the maximum number of iterations was 40. Background levels (Supplementary Table 5) were chosen by manually checking the background of each image. After deconvolution, images were imported in Fiji, adjusted linearly for brightness and contrast, and saved as TIFF files.

For presentation purposes, all images were converted to 8-bit depth using Fiji and arranged into figures using Adobe Illustrator. The schemes presented in the manuscript were made using the BioRender app (BioRender.com) and Adobe Illustrator.

**Click chemistry labeling background intensity measurements**. For the intensity measurements of click chemistry labeling background (data shown in Supplementary Fig. 10), images were acquired on the confocal microscope described above. Data were collected from three independent experiments, 30 images were acquired and analyzed per condition, a total of 180 images for SiR-tz and 180 images for BODIPY-tz labeling. Full 16-bit range images were imported in Fiji, and the background intensity was measured by placing a rectangular ROI in the cell. ROI was placed in the cytoplasm, between neurofilaments, avoiding the nucleus. One ROI was selected per image. Additionally, imaging background was measured by placing the same ROI in the region of the image that contained no cells. Analysis was done blindly, to avoid bias. Parameters measured in ROIs were area, mean intensity, integrated density, and raw integrated density. Raw integrated density was used for calculation of the corrected total cell fluorescence [CTCF = Raw integrated density − (area × mean fluorescence of the imaging background)]. CTCF values were used for the comparison of background values between the conditions.

Analysis of the click labeling background in the presence or absence of eRF1[E55D] (data shown in Supplementary Fig. 16b, c) was done following the same procedure. Transfected cells were identified based on the positive mCherry signal and intensity measurements were done in SiR-tz click labeling channel. A total of 323 images were collected from three independent experiments. Number of cells per group was 96 for NES PylRS/tRNA + eRF1[E55D] + TCO*A-Lys; 58 for NES PylRS/tRNA + eRF1[E55D]; 84 for NES PylRS/tRNA + TCO*A-Lys; 54 for NES PylRS/tRNA; 30 for Non-transfected control + TCO*A-Lys.

**Statistics**. Statistical analyses (Kolmogorov–Smirnov normality test, Kruskal–Wallis test, Mann–Whitney $U$ test, box plots) for background quantifications (data shown in Supplementary Figs. 10 and 16) were carried out with IBM SPSS Statistics Version 25, Armonk, New York, USA. A Kolmogorov–Smirnov test indicated that the data do not follow a normal distribution. Therefore, non-parametric Kruskal–Wallis test was performed. This was followed by a pairwise comparison of groups with Mann–Whitney $U$ tests (with Bonferroni correction for multiple comparisons). Detailed results (mean ranks, $U$, $z$, and $p$ values) of these comparisons are shown in Supplementary Tables 6, 7, and 8.

**Protocol exchange**. A detailed protocol regarding transfections and click chemistry-based labeling of ND7/23 cells and living primary neurons can be accessed here: https://doi.org/10.21203/rs.3.pex-1727/v1[83].

**Reporting summary**. Further information on research design is available in the Nature Research Reporting Summary linked to this article.

## Data availability

All the data supporting the results are provided in the manuscript or supplementary information. Raw data are available on Figshare under the following https://doi.org/10.6084/m9.figshare.c.5749409[80]. We are in the process of depositing plasmids generated in this study on Addgene. Until then, plasmids are available on request (via material transfer agreement free of charge for non-commercial purposes) from the corresponding author. Publicly available dataset Ensembl, release 102, November 2020; Mus musculus version 102.38 (GRCm38.p6; http://nov2020.archive.ensembl.org/Mus_musculus/Info/Index?db=core) was used for the selection of guide sequences for CRISPR/Cas9 genome engineering. Source data are provided with this paper.

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

## Acknowledgements

We would like to thank Katja Widmaier for excellent technical assistance and all the members of the Nikić-Spiegel group for their support, Dr. Adrian Neal for his editorial input, Dr. Rainer Spiegel for advice on statistical analysis, Dr. Edward Lemke and his team for the gift of AF647-tetrazine, TCO4en/eq-Lys and plasmids containing NESPylRS$^{AF}$/tRNA pair and eRF1$^{E55D}$, as well as the laboratories of Dr. Anthony Brown, Dr. Michael Davidson, Dr. Harold MacGillavry, and Dr. Irene Coin for sharing additional plasmids which were obtained through Addgene, as described in the Online Methods. We are also grateful to Dr. Harold MacGillavry and his team for helpful advice about the ORANGE knock-in strategy. The schemes in main and supplementary figures were created with BioRender.com. This study was supported by the Emmy Noether Programme (project number 317530061 to I.N.-S.) of the German Research Foundation (DFG) and the Werner Reichardt Centre for Integrative Neuroscience (Ministry of Science Baden-Württemberg and former Excellence Cluster EXC307 from the DFG). The Leica STED microscope was funded by a grant from the DFG (INST 2388/62-1).

## Author contributions

A.A. designed and performed the experiments involving ND7/23 cells, primary neurons, widefield, and confocal imaging. C.H. performed preliminary experiments involving NFL transfections, mutagenesis, and click labeling in ND7/23 cells. N.S. performed preliminary experiments involving genetic code expansion in ND7/23 cells and helped with image acquisition/data analysis. A.A. and I.N.-S. performed STED imaging, with help from T.S. A.A. and I.N.-S. analyzed the data, prepared the figures, and wrote the manuscript. I.N.-S. conceived and supervised the project. All authors reviewed and approved the final manuscript.

## Competing interests

The authors declare no competing interests.
