## [Peer Review File · Nature Communications]

Reviewers' Comments:

Reviewer #1:

Remarks to the Author:

In this methodological study, the authors show that they can tag a protein in living cultured neurons by combining amber stop codon suppression (to incorporate an unconventional amino acid into the protein at a specific site) with click chemistry (to label the unconventional amino acid with membrane-permeable fluorescent dyes). The method requires the prior identification of mutants that permit efficient codon suppression. Cells are co-transfected with the mutant protein of interest plus a transfer RNA to achieve amber stop codon suppression. Labeling is achieved by adding dye to the medium and subsequently washing out the excess. This approach to protein labeling has been described previously by others. The authors show here that it can also be applied to labeling proteins primary neurons. They also show that the labeling can be achieved by genetic modification of the endogenous protein locus in cultured neurons using CRISPR/Cas9.

To showcase the method, the authors tag neurofilament protein L (NFL). In principle this is a good protein to choose for this purpose because it assembles into discrete polymers that are diffraction-limited in width. The small size of the chemical tags used could be a powerful alternative to genetically encoded fluorescent protein tags. Overall, the study has been executed carefully and the data are of good quality. The mutant protein clearly incorporates into neurofilaments and the authors obtain filamentous labeling with their dye labeling strategy using both confocal and STED imaging.

Specific comments

- The methodology has been applied to other cells types. It is not clear that any particular methodological innovations were required to accomplish this in neurons in the present study.
- The data indicate that the dyes can be prone to both specific and diffuse non-specific background staining resulting in high background fluorescence and reduced SNR, which is a limitation of the approach.
- In the Discussion, the authors state that an advantage of the technique is the potential to achieve higher labeling density but in the images presented the staining of NFL appears weak. For example, it is not clear that the authors are detecting single neurofilament polymers in the confocal or STED images in Fig. 4. Higher magnification would be required to demonstrate this. Single neurofilament polymers are readily detected in live imaging using fluorescent protein tags. It is possible that the authors may be detecting bundles of filaments and that the labeling density is not sufficient to detect single polymers with sufficient signal:noise ratio.
- The authors discuss the challenge of the variable efficiency of amber codon suppression but there is no attempt to quantify that. How is inefficient codon suppression manifested in cells? Could it result in truncated translation products? If so, this could potentially complicate the interpretation of experiments using this approach.
- There is no characterization or quantification of the efficiency of the CRISPR/Cas9 modification so it is unclear how efficient or specific it is.
- In Fig. 3 it is not clear that two different populations of NFL have been labeled because the two labels overlap. This likely reflects the turnover of subunit proteins within these polymers over the course of several days, but it is nonetheless not the best illustration of the potential of the approach. It would be more effective to apply the second label after just a few hours, instead of a few days, which might enable the authors to localize the site of NFL translation in the neuron, for example.
- Fig. 5 seems to be of little utility. The authors state that it demonstrates nanoscale resolution but it is not clear what is being resolved. While the concept of dual-color pulse-chase experiments sounds appealing, it is not clear that it is revealing anything in this experiment.

Reviewer #2:

Remarks to the Author:

This paper illustrates a number of high-quality super resolution images of tagged NFL in neuron cells, via genetic code expansion with unnatural amino acid and bio-orthogonal SPIEDAC between

the incorporated UAA and tetrazine dyes.

NFL has been studied as a biomarker to detect neurodegeneration in brain disorders, thus this minimal genetically encoded tag would be used in advanced studies. Also, authors suggested and demonstrated its usage as in "dual-color pulse-chase labeling of primary neurons," where synthesis of NFL molecules can be monitored through a time scale. By using more than two dyes, visualization of neuronal growth for more short time intervals may be possible as well.

The combination of SPIEDAC labeling with CRISPR/Cas9 genome editing for labeling endogenous protein seems very novel; it has minimal effect on the host cell without overexpression of POI as the authors claimed; also, by introduction of TAG before the 3XFLAG tag, they prevented knockdown of NFL. © Still, this method uses transfection, and the limitations still need to be improved. Also, the issue of protein overexpression can be overcome by the use of weaker promoter. Is there any strong point to the use of CRISPR/Cas9 system?

Though labeling was optimized for living neurons, authors tested four sites in NFL protein for amber suppression. Also, only one out of four sites has shown high expression level and click reaction with low readthrough. To use this technique for labeling other proteins/compartments of neuron, the whole process of selection and optimization need to be done.

Also, the reason why the authors have used the clickable amino acid trans-cyclooct-2-en-L-lysine (TCO*A-Lys) is questionable. TCO*A group was reported to spontaneously cleave off after the SPIEDAC reaction with tetrazine moieties (<https://doi.org/10.1002/anie.201800402>). This may lead to the increased background due to cleaved dye molecules. Why was TCO*A-Lys chosen over other non-cleavable unnatural amino acids such as TCO4en/eq-Lys and endo-BCN-Lys?

The following are some minor issues to be corrected:

1. It could be misunderstood that cells were transfected only with a plasmid conveying NFL gene in fig. 1c and extended data fig. 1a, 1b, 1c, and 1d. In my opinion, +NES PyIRS/tRNA should be additionally described in every case like extended data fig. 1e.
2. I wish there were more explanation about the confocal backgrounds: any methods to resolve it or trials you made. Also, numerical data or graphs about the colocalization would improve the data quality. More colocalization images with anti-NFL antibodies would strengthen your data.
3. Some of the contents in the discussion part should be moved to the introduction part. Especially, the explanation about NFL and the previous researches about the neuronal cells belong to the intro.
4. It was insightful to show that SiR-tz seemed to be co-localized with lysotracker. Yet, in extended data fig. 5b and 5d, we can also see the colocalization of lysotracker and SiR-tz even when UAA isn't added. Isn't there any chance that tetrazine derivatives react with lysotracker?
5. SiR-tz generally seemed to spread over the cells in the data you provided, yet BODIPY-tz seemed to localize well but aggregate some outside of cells. What's the difference between them which causes this phenomenon?
6. Typo : line 232, CRISPR/CAas9  CRISPR/Cas9

Reviewer #3:

Remarks to the Author:

Nikic-Spiegel and coworkers report in this work on combination of genetically encoded unnatural amino acids (UAA) with SPIEDAC labeling in neurons. They focus on neurofilament light chain (NFL) as the target protein. The genetic code expansion (position of the UAA within NFL) was first optimized using rodent ND7/23 cell line. The best mutant was then used to optimize the labeling step where two additional UAAs (TCO, BCN) and different Tz-dyes were investigated. The authors then move on to perform the incorporation/labeling studies in living primary mouse neurons. The

advantage of the presented approach is demonstrated by dual color, pulse-chase labeling experiments. This approach was used to distinguish between distribution of different NFL populations in healthy and injured axons (after oxidative stress). To achieve endogenous levels of NFL, the authors used an elegant combination of the genetic code expansion with CRISPR/Cas9. The manuscript is well written and the claims are well presented and justified. Sufficient experimental details are provided so that others can reproduce the work. Even though no particular advances in the amber suppression/TCO-Tz labeling are presented, the work could be of general interest to the neurobiology community. However, I am not an expert in neurobiology and do not have the required background. I am experienced in incorporation of UAAs and in bioorthogonal click labeling reactions (especially the TCO/Tz chemistry). I am providing my comments from this perspective.

Overall, I recommend publication after the following points will be addressed.

Major points:

1) My first point goes to, in my view, most interesting part of the work where the amber suppression was combined with CRISPR/Cas9 to achieve SPIEDAC labeling of NFL protein at endogenous level.

The authors correctly state in the discussion part (lines 347 – 351) that the amber suppression method may interfere with endogenous translation of other proteins that use the amber codon. When the experiments are performed using transfections/over-expressions, this effect is possibly negligible. However, when proteins are to be studied at endogenous level, this may lead to artifacts. Especially when the engineered eRF1 is used (enhanced amber suppression) as is the case here. To substantiate the results from the CRISPR/Cas9 experiment and to reveal if any background labeling of other proteins is relevant, I suggest to perform the following experiment: transfect MCNs with a pORANGE plasmid encoding for spCas9, gRNA and the NFL-linker-3xFLAG (no-UAG). Incubate the neurons with the TCO* amino acid and perform labeling with Sir-tz under the same conditions as described in Fig. 6. This experiment should reveal if there is any background incorporation of the UAA into other proteins at endogenous level whose translation terminates with the amber codon. If yes, this will show the background which is important to consider for future studies. If there will be no/negligible labeling, the result will substantiate and corroborate the presented results.

2) various Tz dyes were investigated in the study. Some of these probes contain the Me-Tz and some are H-Tz derivatives. The Me-Tz derivatives in combination with the axial isomer of the TCO* lead (at least partially) to cleavage of the TCO moiety after the Diels-Alder reaction in the click-to-release process (work of Mark Robillard and others). This could lead to background labeling due to the release of the click-dye product from the UAA and thus, from the studied protein. This is usually not an issue when labeling is performed on fixed/washed cells, but can influence the experiments performed in live cells. I believe that it is important to point out and comment this in the main text. In fact, some of the background signal presented in supplementary figures 3 and 4 can be a result of this click-to-release process.

3) The authors do not refer to an important, and in some sense similar work of M. Sauer (Communications Biology volume 2, Article number: 261 from 2019) where the SPIEDAC reaction was thoroughly optimized (different Tz dyes, fluorogenicity etc.) for super-resolution microscopy and even combined with the amber suppression (of course not on neurons). This work should be cited and relevance to the presented manuscript shortly discussed and clarified.

Minor points:

1) it is not clear why other UAAs were investigated at the beginning (endo-BCN-Lys and TCOen-Lys). Short comment on the results (advantages/disadvantages) would be helpful and informative to the reader (e.g. no issues with the click-to-release?).

2) There is a mistake in the stereo-chemistry of TCO*A-Lys amino acid structure shown in Fig.1 (the allylic position).

3) There is a mistake in the structure of endo-BCN-Lys structure shown in Extended data Fig. 2. The two bonds on the fused bicyclic system should show hydrogens not methyl groups.

Dear Reviewers,

we are very grateful for your critical reading of our manuscript and all of your comments. We have now performed additional experiments and controls to address these comments. We believe that the review process has allowed us to improve the manuscript and make our claims stronger. These additional experiments, figures (both main and extended data figures) and textual changes include:

- Quantification of the efficiency of amber codon suppression (new experiments were performed and data is shown in new Extended Data Fig. 2)
- Colocalization analysis of click labeling and different antibodies has been performed – both new experiments were performed (new Extended Data Fig. 5) and old images were analyzed (Fig. 2b)
- Experiments with additional dual-color labeling protocols were performed: panels were added to Fig. 3d and new Fig. 4 has been included
- Quantification of the click labeling background – effect of different UAAs and washing procedures (new experiments were performed and data is shown in new Extended Data Fig. 9)
- Efficiency and specificity of ORANGE knock-in strategy (new experiments were performed and data is shown in new Extended Data Fig. 14c)
- Experiments addressing background click labeling with ORANGE knock-in constructs (new experiments were performed and data is shown in new Extended Data Fig. 15a)
- Experiments addressing click labeling background in the presence of eRF1^{E55D} factor (new experiments were performed and data is shown in new Extended Data Fig. 15b,c)
- Additional knock-in method for Targeted Knock-In with Two (TKIT) guides has been established (new experiments were performed and the data is presented in new Extended Data Fig. 16)
- Additional controls for lysosome background – data are shown in new Extended Data Fig. 8
- Textual changes including: discussion of new results, corrections based on reviewers' comments, smaller edits for clarity, etc.

All changes are highlighted in yellow in the main text and supplementary (extended data) information files. Detailed Point-by-Point response is included below.

Reviewer #1

In this methodological study, the authors show that they can tag a protein in living cultured neurons by combining amber stop codon suppression (to incorporate an unconventional amino acid into the protein at a specific site) with click chemistry (to label the unconventional amino acid with membrane-permeable fluorescent dyes). The method requires the prior identification of mutants that permit efficient codon suppression. Cells are co-transfected with the mutant protein of interest plus a transfer RNA to achieve amber stop codon suppression. Labeling is achieved by adding dye to the medium and subsequently washing out the excess. This approach to protein labeling that has been described previously by others. The authors show here that it can also be applied to labeling proteins primary neurons. They also show that the labeling can be achieved by genetic modification of the endogenous protein locus in cultured neurons using CRISPR/Cas9.

To showcase the method, the authors tag neurofilament protein L (NFL). In principle this is a good protein to choose for this purpose because it assembles into discrete polymers that are diffraction-limited in width. The small size of the chemical tags used could be a powerful alternative to genetically encoded

fluorescent protein tags. Overall, the study has been executed carefully and the data are of good quality. The mutant protein clearly incorporates into neurofilaments and the authors obtain filamentous labeling with their dye labeling strategy using both confocal and STED imaging.

We thank the reviewer for their positive feedback.

Specific comments

- *The methodology has been applied to other cells types. It is not clear that any particular methodological innovations were required to accomplish this in neurons in the present study.*

Indeed, as we also discuss in our manuscript, the technology itself is not novel. However, considering that neurons are not easy-to-transfect standard cell lines, bringing this technology into neurons represented a significant challenge that was unachievable for quite some time. It was not clear if the expression levels of TAG mutants will be high enough and if the orthogonal translational machinery will be efficient enough. For this, different parameters had to be optimized. Furthermore, what we show here for the first time is that live-cell click labeling can be combined with genome engineering. We demonstrate this in primary neurons, but the method can also be applied in any other type of cells. Last but not least, in order to make it easier for others to establish and use this technology, we provide a workflow for establishing click chemistry-based labeling and we openly discuss advantages and limitations of this method. We believe that all of this together will be very valuable for the members of the neuroscience community. Recent reviews in the field (for example, <https://doi.org/10.1038/s41583-021-00441-z> and <https://doi.org/10.1038/s41593-019-0480-6>) bring up the topic of click labeling as one of the emerging methods for microscopy studies. However, we believe that despite the excitement about the possibilities of this method, it is important to keep a critical point of view. Being one of the pioneers in the field of click-based protein labeling for advanced microscopy studies, we are more than aware of both limitations and advantages. Consequently, we discussed these in detail in our manuscript.

- *The data indicate that the dyes can be prone to both specific and diffuse non-specific background staining resulting in high background fluorescence and reduced SNR, which is a limitation of the approach.*

This is another point that we agree on. Click labeling, especially of intracellular proteins, does not come without limitations. One of them is the diffuse non-specific background signal. Prompted by the questions of all three reviewers, we have now performed additional experiments and analyses to address different factors that contribute to the background. The results are presented in Extended Data Figure 9 (UAA- and washing procedures-specific background) and Extended data fig 15 (eRF1^{E55D}-specific background). Furthermore, we have emphasized this limitation in the discussion (lines 474 to 494).

- *In the Discussion, the authors state that an advantage of the technique is the potential to achieve higher labeling density but in the images presented the staining of NFL appears weak. For example, it is not clear that the authors are detecting single neurofilament polymers in the confocal or STED images in Fig. 4. Higher magnification would be required to demonstrate this. Single neurofilament polymers are readily detected in live imaging using fluorescent protein tags. It is possible that the authors may be detecting bundles of filaments and that the labeling density is not sufficient to detect single polymers with sufficient signal:noise ratio.*

We are not certain if we fully understand this remark, but we will try to address it here. What we state in the manuscript discussion is based on the theoretical assumption. Due to their size, smaller labels can be put at higher density than larger labels (such as antibodies). In addition, click labeling is stoichiometric (tetrazine dye binds the UAA at 1:1 ratio). Organic dye-conjugated antibodies are frequently used in super-resolution microscopy but they are rather large, they introduce “linkage” errors and the labeling is not stoichiometric; this is the comparison that we had in mind. While fluorescent protein tags can also be considered stoichiometric, due to different photophysical properties of proteins and organic dyes they cannot be directly compared. Unfortunately, we have no way to quantify if the labeling density is indeed high(er) in the case of NFL and we did not make any comparisons or claims in this regard. On the other hand, if the labeling density was not high enough we would expect more punctae-like staining. For example, STED images of immunostained neurofilament medium chain in Figure 4 of this manuscript <https://doi.org/10.1111/j.1365-2818.2009.03188.x> look different than what we see in our images. It is also important to note that with STED microscopy we cannot resolve individual filaments that have 10nm diameter. There is an improvement in the resolution compared to the confocal, but conventional STED microscopy does not allow us to go below 10nm limit, so what we see are bundles of filaments. In the previous manuscript version, we referred to these as NFL-fibers. We have now corrected the main text by replacing “NFL-fibers” with NFL bundles (line 261). Considering the technology which is available in most laboratories, we would roughly estimate the achievable lateral resolution of STORM and STED to 30nm and 70nm, respectively. With such resolution, single polymers cannot be detected regardless of the labeling method. We do not know if we overlooked any specific manuscript that the reviewer might have had in mind, but the resolution of conventional microscopy is in any case lower than confocal or STED microscopy. To resolve individual polymers within a dense neurofilament network higher resolution would be needed.

- *The authors discuss the challenge of the variable efficiency of amber codon suppression but there is no attempt to quantify that. How is inefficient codon suppression manifested in cells? Could it result in truncated translation products? If so, this could potentially complicate the interpretation of experiments using this approach.*

We thank the reviewer for this comment. We have performed new experiments to quantify the efficiency of amber codon suppression for NFL^{TAG} mutants. For the most efficiently expressing mutant NFL^{K363TAG} we get around 40% compared to the expression levels of NFL^{WT}. The data is presented in Extended Data Fig. 2. On the other hand, truncated proteins will not contain UAA and thus cannot be fluorescently labeled and cannot complicate the interpretation of microscopy imaging results.

- *There is no characterization or quantification of the efficiency of the CRISP/Cas9 modification so it is unclear how efficient or specific it is.*

We have performed new experiments to quantify both the efficiency and specificity of CRISPR/Cas9 modifications. As one of the limitations of ORANGE genome editing was the change in the *Nefl* ORF (due to the limited number of suitable protospacer adjacent motif (PAM) sites at the end of the *Nefl* gene), we have also introduced new approach based on targeted knock-in with two guides (TKIT). Analogously to the ORANGE, we have performed the quantification of efficiency and specificity of TKIT-based knock-in. These results are introduced in the text and shown in the Extended Data Fig. 14 and 16.

- *In Fig. 3 it is not clear that two different populations of NFL have been labeled because the two labels overlap. This likely reflects the turnover of subunit proteins within these polymers over the course of several days, but it is nonetheless not the best illustration of the potential of the approach. It would be*

more effective to apply the second label after just a few hours, instead of a few days, which might enable the authors to localize the site of NFL translation in the neuron, for example.

We thank the reviewer for this suggestion. The versatility of genetic code expansion and SPIEDAC labeling is beautifully illustrated by dual-color labeling and it is one of its great advantages. However, we agree that the initial approach for dual-color labeling was not the best example of the potential of the method. We have now performed two different types of experiments. In one, following the reviewer's suggestion, we labeled the 2nd population 3 h after the first click reaction. This confirmed that NFL subunits are incorporated very fast into the existing neurofilament network. In some instances, we were even able to identify chimeric neurofilaments that contained distinct segments labeled with two tetrazine dyes (Fig. 3d). In the second approach, we waited longer and introduced a UAA-free "pause" during the NFL^{TAG} translation – this allowed us to spatially separate two populations in a more obvious way (Figure 4).

- *Fig. 5 seems to be of little utility. The authors state that it demonstrates nanoscale resolution but it is not clear what is being resolved. While the concept of dual-color pulse-chase experiments sounds appealing, it is not clear that it is revealing anything in this experiment.*

We now provide higher magnification to illustrate organization of NFL in axonal swellings and we moved the figure to the supplement. We think that this is still relevant since it was not clear if the signal will be high enough to resolve NFL in axons in two colors with STED microscopy. For this reason, we would still like to keep this as one of the Extended Data figures. We hope that the reviewer will agree with us.

Reviewer #2

This paper illustrates a number of high-quality super resolution images of tagged NFL in neuron cells, via genetic code expansion with unnatural amino acid and bio-orthogonal SPIEDAC between the incorporated UAA and tetrazine dyes.

NFL has been studied as a biomarker to detect neurodegeneration in brain disorders, thus this minimal genetically encoded tag would be used in advanced studies. Also, authors suggested and demonstrated its usage as in "dual-color pulse-chase labeling of primary neurons," where synthesis of NFL molecules can be monitored through a time scale. By using more than two dyes, visualization of neuronal growth for more short time intervals may be possible as well.

We thank the reviewer for their positive feedback.

- *The combination of SPIEDAC labeling with CRISPR/Cas9 genome editing for labeling endogenous protein seems very novel; it has minimal effect on the host cell without overexpression of POI as the authors claimed; also, by introduction of TAG before the 3XFLAG tag, they prevented knockdown of NFL. 😊 Still, this method uses transfection, and the limitations still need to be improved. Also, the issue of protein overexpression can be overcome by the use of weaker promoter. Is there any strong point to the use of CRISPR/Cas9 system?*

This question does not have a simple yes or no answer. The answer will depend on the experimental design and biological question of interest. Both overexpression/transfection and endogenous protein tagging have their advantages and disadvantages. As for weaker promoters, they could also be used. But this strategy would also need to be tested for each of the proteins of interest. In general, as with other

genetically encoded tags, one needs to find a balance between achieving enough protein expression to get good signal to noise ratio and not perturbing the protein's function. And we would rather say 😊 to the solution of the problem with the linker strategy.

- *Though labeling was optimized for living neurons, authors tested four sites in NFL protein for amber suppression. Also, only one out of four sites has shown high expression level and click reaction with low readthrough. To use this technique for labeling other proteins/compartments of neuron, the whole process of selection and optimization need to be done.*

Yes, that is true. Like with other genetically encoded tags (including fluorescent protein fusions, self-labeling tags etc.), each protein of interest is a new experiment. Click labeling is a bit more complex since it involves selection of permissive sites, but we provide a workflow and enough experimental information that will help others establish genetic code expansion and click labeling for their proteins of interest.

- *Also, the reason why the authors have used the clickable amino acid trans-cyclooct-2-en-L-lysine (TCO*A-Lys) is questionable. TCO*A group was reported to spontaneously cleave off after the SPIEDAC reaction with tetrazine moieties (<https://doi.org/10.1002/anie.201800402>). This may lead to the increased background due to cleaved dye molecules. Why was TCO*A-Lys chosen over other non-cleavable unnatural amino acids such as TCO4en/eq-Lys and endo-BCN-Lys?*

We thank the reviewer for bringing this up. This is a very important point that we clarified in the revised manuscript. As we also discuss in the revised manuscript, we have previously shown superiority of a mixture of isomers of TCO*-Lys over BCN-Lys for the labeling and microscopy studies of extracellular proteins (<https://doi.org/10.1038/nprot.2015.045>). In this manuscript, we used a more stable isoform, axial TCO*A-Lys (<https://doi.org/10.1002/chem.201501647>). In addition, most of the used dyes are unsubstituted H-tetrazines so the elimination reaction is less of a concern. However, to be on the safe side and because of a) the potential problems with unwanted cleavage/elimination reaction of the click-product and b) previous work using endo-BCN-Lys for intracellular labeling (<https://doi.org/10.1021/ja512838z>, <https://doi.org/10.1091/mbc.e17-03-0161>), we also tested TCO4en/eq-Lys and endo-BCN-Lys. However, we had problems with their incorporation in neurons. TCO4en/eq incorporation was very low. Endo-BCN-Lys induced cytotoxicity during 3 day-incubation periods. During the revision, we readdressed this problem and tried to solve it. By incubating neurons shorter (2 days vs. 3 days) we could get expression with endo-BCN-Lys. We still believe that endo-BCN-Lys is a good alternative, but its incorporation in neurons will need to be optimized. This could also be a batch-specific problem that we did not have the chance to address until now. As for TCO4en/eq-Lys, different PylRS variant will need to be used. We discuss this in the revised manuscript (lines 138-143 and 477-490).

The following are some minor issues to be corrected:

1. *It could be misunderstood that cells were transfected only with a plasmid conveying NFL gene in fig. 1c and extended data fig. 1a, 1b, 1c, and 1d. In my opinion, +NES PylRS/tRNA should be additionally described in every case like extended data fig. 1e.*

Corresponding labels have been added to the figures.

2. I wish there were more explanation about the confocal backgrounds: any methods to resolve it or trials you made. Also, numerical data or graphs about the colocalization would improve the data quality. More colocalization images with anti-NFL antibodies would strengthen your data.

We thank the reviewer for bringing this up. We have performed new experiments to quantify different sources of the background signal. New experiments were needed in order to have proper comparisons of different factors. In this regard, we compared different UAAs (TCO*A-Lys and endo-BCN-Lys), different dyes (SiR-tetrazine and BODIPY-tetrazine) and washing protocols. Data are shown in the Extended Data Fig. 9 and discussed in the revised text. We also performed colocalization analysis with old and new images (Figure 2 and Extended Data Fig. 5). New experiments were performed after we noticed that anti-NFL/SiR-tz colocalization was better than anti-FLAG/SiR-tz colocalization. In a number of images, anti-FLAG was not as strong in the center of the cell body as on the periphery (also visible in Fig 1d). We first thought that this discrepancy is related to the immunostaining procedure. It turned out to be related to the anti-FLAG antibody that we used. When we changed the primary anti-FLAG antibody the colocalization was better (Extended Data Fig. 5).

3. Some of the contents in the discussion part should be moved to the introduction part. Especially, the explanation about NFL and the previous researches about the neuronal cells belong to the intro.

We have tried to fix this by text edits. Explanation about NFL is in the revised introduction (lines 89-97). We have also tried to incorporate the part about genetic code expansion experiments in neurons in the introduction. However, for clarity reasons, we think that keeping the focus of introduction on fluorescent labeling and contributions from the field of click chemistry to this aspect is more appropriate. In the discussion, we then put our results into a more general context of the genetic code expansion in neurobiology. We hope that the reviewer will find these edits suitable.

4. It was insightful to show that SiR-tz seemed to be co-localized with lysotracker. Yet, in extended data fig. 5b and 5d, we can also see the colocalization of lysotracker and SiR-tz even when UAA isn't added. Isn't there any chance that tetrazine derivatives react with lysotracker?

If we understand this remark correctly, the reviewer thinks that the tetrazine group could react with the lysotracker. Here we have to emphasize that we also saw SiR-tz (or other tetrazine dyes) signal in lysosomes in the absence of lysotracker. That is also how we noticed it in the first place (Figure 2c). However, to be sure about this, we performed additional control experiments. We did click labeling with two dyes (SiR-tz and BODIPY-tz) in the presence and absence of lysotracker. No difference was observed in the tendency of tetrazine dyes to accumulate in lysosomes. The data is shown in Extended Data Fig. 8.

5. SiR-tz generally seemed to spread over the cells in the data you provided, yet BODIPY-tz seemed to localize well but aggregate some outside of cells. What's the difference between them which causes this phenomenon?

The aggregates of BODIPY-tz shown in the Figure 2 most likely come from the accumulation of BODIPY-tz in cell debris. We do not think that this is specific for BODIPY-tz. When we did the comparison of click labeling background (Extended Data Fig 9), we took a number of images with both SiR-tz and BODIPY-tz in a comparative way. Depending on the field of view, such accumulations were present and both dyes show same tendency to form them.

6. Typo : line 232, CRISPR/CAas9  CRISPR/Cas9

Typo has been fixed.

Reviewer #3 (Remarks to the Author):

Nikic-Spiegel and coworkers report in this work on combination of genetically encoded unnatural amino acids (UAA) with SPIEDAC labeling in neurons. They focus on neurofilament light chain (NFL) as the target protein. The genetic code expansion (position of the UAA within NFL) was first optimized using rodent ND7/23 cell line. The best mutant was then used to optimize the labeling step where two additional UAAs (TCO, BCN) and different Tz-dyes were investigated. The authors then move on to perform the incorporation/labeling studies in living primary mouse neurons. The advantage of the presented approach is demonstrated by dual color, pulse-chase labeling experiments. This approach was used to distinguish between distribution of different NFL populations in healthy and injured axons (after oxidative stress). To achieve endogenous levels of NFL, the authors used an elegant combination of the genetic code expansion with CRISPR/Cas9.

The manuscript is well written and the claims are well presented and justified. Sufficient experimental details are provided so that others can reproduce the work. Even though no particular advances in the amber suppression/TCO-Tz labeling are presented, the work could be of general interest to the neurobiology community. However, I am not an expert in neurobiology and do not have the required background. I am experienced in incorporation of UAAs and in bioorthogonal click labeling reactions (especially the TCO/Tz chemistry). I am providing my comments from this perspective. Overall, I recommend publication after the following points will be addressed.

We thank the reviewer for their positive feedback.

Major points:

1) My first point goes to, in my view, most interesting part of the work where the amber suppression was combined with CRISPR/Cas9 to achieve SPIEDAC labeling of NFL protein at endogenous level. The authors correctly state in the discussion part (lines 347 – 351) that the amber suppression method may interfere with endogenous translation of other proteins that use the amber codon. When the experiments are performed using transfections/over-expressions, this effect is possibly negligible. However, when proteins are to be studied at endogenous level, this may lead to artifacts. Especially when the engineered eRF1 is used (enhanced amber suppression) as is the case here. To substantiate the results from the CRISPR/Cas9 experiment and to reveal if any background labeling of other proteins is relevant, I suggest to perform the following experiment: transfect MCNs with a pORANGE plasmid encoding for spCas9, gRNA and the NFL-linker-3xFLAG (no-UAG). Incubate the neurons with the TCO* amino acid and perform labeling with Sir-tz under the same conditions as described in Fig. 6. This experiment should reveal if there is any background incorporation of the UAA into other proteins at endogenous level whose translation terminates with the amber codon. If yes, this will show the background which is important to consider for future studies. If there will be no/negligible labeling, the result will substantiate and corroborate the presented results.

We thank the reviewer for bringing this up. We have performed suggested experiments and we can see that the specific labeling is above the background level. The results are shown in Extended Data Fig. 15a. We did the same for the TKIT genome engineering technology that we introduced in the revised manuscript. The reason for introducing TKIT technology is that the ORANGE technology led to the

change in the *Nefl* ORF (due to the limited number of suitable protospacer adjacent motif sites at the end of the *Nefl* gene). The TKIT data with all the relevant controls is shown in Extended Data Fig. 16. However, as click labeling background due to endogenous amber codon suppression remains a very important issue, we discuss this more extensively in the revised manuscript (e.g. lines 303 – 313). In addition, to exclude if eRF1^{E55D} alone contributes to the higher click labeling background due to increased rate of amber codon suppression, we performed additional control experiments in which we directly compared levels of background with and without eRF1^{E55D}. The results are shown in Extended Data Fig. 15b,c and discussed in lines 303-323; 502 – 507.

2) various Tz dyes were investigated in the study. Some of these probes contain the Me-Tz and some are H-Tz derivatives. The Me-Tz derivatives in combination with the axial isomer of the TCO lead (at least partially) to cleavage of the TCO moiety after the Diels-Alder reaction in the click-to-release process (work of Mark Robillard and others). This could lead to background labeling due to the release of the click-dye product from the UAA and thus, from the studied protein. This is usually not an issue when labeling is performed on fixed/washed cells, but can influence the experiments performed in live cells. I believe that it is important to point out and comment this in the main text. In fact, some of the background signal presented in supplementary figures 3 and 4 can be a result of this click-to-release process.*

We thank the reviewer for bringing this up. This is a very important point that was also brought up by reviewer 2 and that we clarified in the revised manuscript (e.g. lines 138-143).

3) The authors do not refer to an important, and in some sense similar work of M. Sauer (Communications Biology volume 2, Article number: 261 from 2019) where the SPIEDAC reaction was thoroughly optimized (different Tz dyes, fluorogenicity etc.) for super-resolution microscopy and even combined with the amber suppression (of course not on neurons). This work should be cited and relevance to the presented manuscript shortly discussed and clarified.

We have included discussion of this work in the revised manuscript. As we also discuss in the manuscript (lines 439-453), we would like to point out some important differences. Beliu et al. provide a very thorough comparison of spectroscopic and fluorogenic properties of a number of tetrazine dyes. This also allowed them to identify a new mechanism for fluorogenicity. However, comparative tetrazine labeling experiments were done post-fixation with phalloidin-TCO. As this type of labeling is different than live-cell click labeling of intracellular proteins bearing UAAs in the presence of genetic code expansion machinery, some of the dyes might behave differently under more complex conditions of genetic code expansion. The authors do not comment on this but the fact that only a subset of already known tetrazine dyes (SiR, HMSIR, Cy5) was subsequently used for site-specific labeling of proteins bearing TCO*A-Lys left the question about optimal labeling conditions with different tetrazine dyes open.

Minor points:

1) it is not clear why other UAAs were investigated at the beginning (endo-BCN-Lys and TCOen-Lys). Short comment on the results (advantages/disadvantages) would be helpful and informative to the reader (e.g. no issues with the click-to-release?).

This is a very important question and since the same question was asked by the reviewer 2, we will copy our answer to them here. As we also discuss in the revised manuscript, we have previously shown superiority of a mixture of isomers of TCO*-Lys over BCN-Lys for the labeling and microscopy studies of

extracellular proteins (<https://doi.org/10.1038/nprot.2015.045>). In this manuscript, we used a more stable isoform, axial TCO*A-Lys (<https://doi.org/10.1002/chem.201501647>). In addition, most of the used dyes are unsubstituted H-tetrazines so the elimination reaction is less of a concern. However, to be on the safe side and because of a) the potential problems with unwanted cleavage/elimination reaction of the click-product and b) previous work using endo-BCN-Lys for intracellular labeling (<https://doi.org/10.1021/ja512838z>, <https://doi.org/10.1091/mbc.e17-03-0161>), we also tested TCO4en/eq-Lys and endo-BCN-Lys. However, we had problems with their incorporation in neurons. TCO4en/eq incorporation was very low. Endo-BCN-Lys induced cytotoxicity during 3 day-incubation periods. During the revision, we readdressed this problem and tried to solve it. By incubating neurons shorter (2 days vs. 3 days) we could get expression with endo-BCN-Lys. We still believe that endo-BCN-Lys is a good alternative, but its incorporation in neurons will need to be optimized. This could also be a batch-specific problem that we did not have the chance to address until now. As for TCO4en/eq-Lys, different PylRS variant will need to be used. We discuss this in the revised manuscript (lines 138-143 and 477-490).

*2) There is a mistake in the stereo-chemistry of TCO*A-Lys amino acid structure shown in Fig.1 (the allylic position).*

We have corrected the structure. Considering complex stereochemistry of TCO*A-Lys, a simplified 2D representation is included in the revised Figure 2b.

3) There is a mistake in the structure of endo-BCN-Lys structure shown in Extended data Fig. 2. The two bonds on the fused bicyclic system should show hydrogens not methyl groups.

We have corrected this mistake.

Reviewers' Comments:

Reviewer #1:

Remarks to the Author:

The authors have done a thorough job of responding to my critiques and comments and have made extensive changes to the manuscript. Below are my specific comments on the authors' responses. The edits to the text and figures and the additional supplementary data have improved the manuscript, but some important concerns have not been addressed. In particular, the authors have not adequately addressed the concern that the low efficiency of amber codon suppression may result in the expression of truncated protein, or in the case of their CRISPR/TKIT approach, that it may result in partial knockdown of the targeted protein. The goal of the authors' method is to probe cellular function with minimal perturbation. If their method introduces unlabeled truncated proteins or reduces protein levels, this could seriously undermine this goal. I feel that this is a fundamental concern, and should be addressed both experimentally and in the discussion in the manuscript.

MY ORIGINAL COMMENT: The methodology has been applied to other cell types. It is not clear that any particular methodological innovations were required to accomplish this in neurons in the present study.

MY RESPONSE TO THE REVISED MANUSCRIPT: The authors have explained what is novel about their approach. I agree and am satisfied with their response.

MY ORIGINAL COMMENT: The data indicate that the dyes can be prone to both specific and diffuse non-specific background staining resulting in high background fluorescence and reduced SNR, which is a limitation of the approach.

MY RESPONSE TO THE REVISED MANUSCRIPT: The authors have added supplementary data which demonstrate that lower background staining can be obtained using endo-BCN-Lys reagent, though this would appear to be of limited utility for studies on neurons because of its toxicity. The authors have added a discuss this limitation, which I feel is sufficient. Thus, I am satisfied with their response.

MY ORIGINAL COMMENT: In the Discussion, the authors state that an advantage of the technique is the potential to achieve higher labeling density but in the images presented the staining of NFL appears weak. For example, it is not clear that the authors are detecting single neurofilament polymers in the confocal or STED images in Fig. 4. Higher magnification would be required to demonstrate this. Single neurofilament polymers are readily detected in live imaging using fluorescent protein tags. It is possible that the authors may be detecting bundles of filaments and that the labeling density is not sufficient to detect single polymers with sufficient signal:noise ratio.

MY RESPONSE TO THE REVISED MANUSCRIPT: I apologize if my point was not clear. There may be some confusion regarding the terms resolution and detection. The authors are correct that single neurofilament polymers cannot be resolved if they are closer together than the optical resolution limit. However, single neurofilaments can be detected when they are a sufficiently low density. There are many papers that demonstrate this, both by live and immunofluorescence imaging. For example, ref. 42 which they cite in the manuscript. My concern was that the authors tout the high labeling density obtained with their method, yet it is not possible to verify that if we are looking at bundles of neurofilaments. A more stringent test would be to image single diffraction-limited neurofilament polymers. If the fluorescence along single neurofilament polymers is continuous, as it often is in experiments using fluorescent fusion proteins, then this would support the authors' assertion. Inspection of Figs. 3 and 5 in the revised manuscript does suggest to me that single NFs are being detected, but it is not clear to me that the labeling density is high relative to data obtained by others using fluorescent fusion proteins. This should be discussed.

MY ORIGINAL COMMENT: The authors discuss the challenge of the variable efficiency of amber codon suppression but there is no attempt to quantify that. How is inefficient codon suppression manifested in cells? Could it result in truncated translation products? If so, this could potentially complicate the interpretation of experiments using this approach.

MY RESPONSE TO THE REVISED MANUSCRIPT: In the revised manuscript, the authors show that the most efficient amber codon suppression obtained was 40%. In their rebuttal, the authors write "On the other hand, truncated proteins will not contain UAA and thus cannot be fluorescently labeled and cannot complicate the interpretation of microscopy imaging results." While the authors are correct that truncated proteins will not be labeled, they do not appear to consider the possibility that the truncated proteins could perturb cellular function. Just because we do not see the truncated proteins does not mean they are not a problem! This could be a serious limitation in terms of the general utility of their approach. The authors do Western blotting of the expressed proteins in Extended Data Fig. 2, but they use a C-terminal FLAG tag on the NFL and probe with a FLAG antibody so they only see the full-length NFL protein. The truncated proteins may or may not be stable, but this experiment does not appear to address this. For example, they obtain the best suppression with the K363 mutant but K363 lies within coil 2 of the NFL rod domain. If stably expressed, fragments of NFL truncated in this region of the rod domain could perturb neurofilament assembly in a dominant manner. Clearly the authors are not seeing wholesale disruption of neurofilament assembly in their data because they do see tagged NFL protein incorporation into neurofilaments, but can they exclude some disruption? Moreover, even if this is not an issue for NFL, it might be an issue for other proteins and thus limit the more general applicability of the methodology. The authors should discuss this in the manuscript, and they should perform the experiment in Extended Data Fig2 using an N-terminal FLAG tag to confirm the presence or absence of truncated protein fragments arising from the 60% of transcripts for which there is no amber codon suppression.

MY ORIGINAL COMMENT: There is no characterization or quantification of the efficiency of the CRISP/Cas9 modification, so it is unclear how efficient or specific it is.

MY RESPONSE TO THE REVISED MANUSCRIPT: One line 286 of the revised manuscript the authors write "Because of the limited efficiency of amber codon suppression, site-specific introduction of TAG at the position K363 would result in knock-down of the endogenous NFL. Therefore, we changed the approach by adding a hexapeptide linker and a 3xFLAG tag at the C terminus of NFL (Fig. 6a)." The use of the C-terminal tag ensures that the authors will only see full-length protein, but it does not solve the potential problem of partial knockdown of NFL in these cells. The authors have not shown what the effects are on the overall level of NFL in the cells. If the labeling is accompanied by partial knockdown, this would limit the general utility of the authors' approach.

On line 292 of the revised manuscript, the authors write "The specificity was also confirmed by sequencing of the PCR product flanking the integration sites in genomic DNA isolated from transfected neurons. The efficiency of the 293 knock-in 6 days after transfection is around 20% (Extended Data Fig. 14c)." The sequencing data is not shown so the specificity is still unclear. However, this is a lesser concern for me as the specificity of CRISPR is not central to this study.

MY ORIGINAL COMMENT: In Fig. 3 it is not clear that two different populations of NFL have been labeled because the two labels overlap. This likely reflects the turnover of subunit proteins within these polymers over the course of several days, but it is nonetheless not the best illustration of the potential of the approach. It would be more effective to apply the second label after just a few hours, instead of a few days, which might enable the authors to localize the site of NFL translation in the neuron, for example.

MY RESPONSE TO THE REVISED MANUSCRIPT: The authors have performed additional experiments and added new data to address this suggestion. I think it is a much better illustration of the utility of their approach.

MY ORIGINAL COMMENT: Fig. 5 seems to be of little utility. The authors state that it demonstrates nanoscale resolution but it is not clear what is being resolved. While the concept of dual-color pulse-chase experiments sounds appealing, it is not clear that it is revealing anything in this experiment.

MY RESPONSE TO THE REVISED MANUSCRIPT: The authors moved the figure to the supplement and increased the magnification in some of the panels. I am fine with this.

Reviewer #3:

Remarks to the Author:

The additional experiments and corrections made in the revised manuscript address my previous concerns. In my view, the manuscript has improved considerably and all issues were sufficiently clarified in the point-by-point response to reviewer comments. I do not have additional comments.

Dear Reviewers,

Dear Editor,

we are very grateful for your critical reading of the revised manuscript and all of your comments. We have now performed additional experiments to address these comments. We believe that the 2nd round of the review process has allowed us to further improve the manuscript.

All changes are highlighted in turquoise in the main text and supplementary (extended data) information files. Detailed Point-by-Point response is included below.

Reviewer #1 (Remarks to the Author):

The authors have done a thorough job of responding to my critiques and comments and have made extensive changes to the manuscript. Below are my specific comments on the authors' responses. The edits to the text and figures and the additional supplementary data have improved the manuscript, but some important concerns have not been addressed. In particular, the authors have not adequately addressed the concern that the low efficiency of amber codon suppression may result in the expression of truncated protein, or in the case of their CRISPR/TKIT approach, that it may result in partial knockdown of the targeted protein. The goal of the authors' method is to probe cellular function with minimal perturbation. If their method introduces unlabeled truncated proteins or reduces protein levels, this could seriously undermine this goal. I feel that this is a fundamental concern, and should be addressed both experimentally and in the discussion in the manuscript.

We thank the reviewer for these comments. We have now performed additional experiments that directly address the concern related to the expression of truncated proteins. While this problem is intrinsic to the amber codon suppression technology (not only in primary neurons but any type of cells or multicellular organisms) and it is challenging to completely avoid it when using the plasmid-based overexpression of the protein of interest, it is important to point out that our CRISPR/Cas9 strategy will not reduce target protein expression levels. We discuss this in detail in the corresponding sections below and in the revised manuscript.

MY ORIGINAL COMMENT: The methodology has been applied to other cell types. It is not clear that any particular methodological innovations were required to accomplish this in neurons in the present study.

MY RESPONSE TO THE REVISED MANUSCRIPT: The authors have explained what is novel about their approach. I agree and am satisfied with their response.

We thank the reviewer for this comment.

MY ORIGINAL COMMENT: The data indicate that the dyes can be prone to both specific and diffuse non-specific background staining resulting in high background fluorescence and reduced SNR, which is a limitation of the approach.

MY RESPONSE TO THE REVISED MANUSCRIPT: The authors have added supplementary data which demonstrate that lower background staining can be obtained using endo-BCN-Lys reagent, though this would appear to be of limited utility for studies on neurons because of its toxicity. The authors have added a discuss this limitation, which I feel is sufficient. Thus, I am satisfied with their response.

We thank the reviewer for this comment.

MY ORIGINAL COMMENT: In the Discussion, the authors state that an advantage of the technique is the potential to achieve higher labeling density but in the images presented the staining of NFL appears weak. For example, it is not clear that the authors are detecting single neurofilament polymers in the confocal or STED images in Fig. 4. Higher magnification would be required to demonstrate this. Single neurofilament polymers are readily detected in live imaging using fluorescent protein tags. It is possible that the authors may be detecting bundles of filaments and that the labeling density is not sufficient to detect single polymers with sufficient signal:noise ratio.

MY RESPONSE TO THE REVISED MANUSCRIPT: I apologize if my point was not clear. There may be some confusion regarding the terms resolution and detection. The authors are correct that single neurofilament polymers cannot be resolved if they are closer together than the optical resolution limit. However, single neurofilaments can be detected when they are a sufficiently low density. There are many papers that demonstrate this, both by live and immunofluorescence imaging. For example, ref. 42 which they cite in the manuscript. My concern was that the authors tout the high labeling density obtained with their method, yet it is not possible to verify that if we are looking at bundles of neurofilaments. A more stringent test would be to image single diffraction-limited neurofilament polymers. If the fluorescence along single neurofilament polymers is continuous, as it often is in experiments using fluorescent fusion proteins, then this would support the authors' assertion. Inspection of Figs. 3 and 5 in the revised manuscript does suggest to me that single NFLs are being detected, but it is not clear to me that the labeling density is high relative to data obtained by others using fluorescent fusion proteins. This should be discussed.

We agree with the reviewer that the labeling density of click labelled NFL might not be higher than the data obtained using fluorescent protein fusions. The corresponding section (which we now modified to avoid confusion) in the discussion was not implying that click-labeled NFL has higher labeling density than GFP-labeled NFL. This part of the discussion is referring to the protein labeling methods that are used for super-resolution imaging (of mainly fixed cells) and the requirement to reduce the "linkage error" as well as to increase the labeling density in order to achieve optimal resolution. In line with what we also wrote in our previous rebuttal letter, we now specify that the higher labeling density is based on the theoretical assumption and previous work in the field of super-resolution microscopy which compares conventional antibody-based labeling to smaller labeling tags, such as nanobodies, aptamers etc. In this regard, we also discuss previous work which directly compared antibodies and click-based site-specific labeling of NMDA receptors. It is also important to note that in contrast to the antibody-based labeling, both GFP-fusions and click labeling are stoichiometric (we expect 1:1 ratio of GFP to NFL,

as well as 1:1 ratio of the tetrazine dye to NFL). However, the practically achievable labeling densities might differ from the densities that are based on the theoretical assumption. We have now expanded the corresponding discussion section to include all aforementioned points (lines 438-450 in the revised manuscript).

MY ORIGINAL COMMENT: The authors discuss the challenge of the variable efficiency of amber codon suppression but there is no attempt to quantify that. How is inefficient codon suppression manifested in cells? Could it result in truncated translation products? If so, this could potentially complicate the interpretation of experiments using this approach.

MY RESPONSE TO THE REVISED MANUSCRIPT: In the revised manuscript, the authors show that the most efficient amber codon suppression obtained was 40%. In their rebuttal, the authors write “On the other hand, truncated proteins will not contain UAA and thus cannot be fluorescently labeled and cannot complicate the interpretation of microscopy imaging results.” While the authors are correct that truncated proteins will not be labeled, they do not appear to consider the possibility that the truncated proteins could perturb cellular function. Just because we do not see the truncated proteins does not mean they are not a problem! This could be a serious limitation in terms of the general utility of their approach. The authors do Western blotting of the expressed proteins in Extended Data Fig. 2, but they use a C-terminal FLAG tag on the NFL and probe with a FLAG antibody so they only see the full-length NFL protein. The truncated proteins may or may not be stable, but this experiment does not appear to address this. For example, they obtain the best suppression with the K363 mutant but K363 lies within coil 2 of the NFL rod domain. If stably expressed, fragments of NFL truncated in this region of the rod domain could perturb neurofilament assembly in a dominant manner. Clearly the authors are not seeing wholesale disruption of neurofilament assembly in their data because they do see tagged NFL protein incorporation into neurofilaments, but can they exclude some disruption? Moreover, even if this is not an issue for NFL, it might be an issue for other proteins and thus limit the more general applicability of the methodology. The authors should discuss this in the manuscript, and they should perform the experiment in Extended Data Fig2 using an N-terminal FLAG tag to confirm the presence or absence of truncated protein fragments arising from the 60% of transcripts for which there is no amber codon suppression.

We apologize for the confusion. We at first referred to the remarks related to the background signal which was in the focus of the first round of revision. And we wanted to point out that truncated proteins will not contribute to the fluorescence background. We agree that this does not make them disappear and share the concern that they might interfere with neurofilament assembly or influence overall cell health in some other way. We appreciate the detailed response of the reviewer in this round of revision which made us realize our overlook. To understand to what extent truncated fragments are present, we performed additional experiments. Following the reviewer’s suggestion, we cloned new constructs containing FLAG-tag at the N-terminus of NFL^{WT} and NFL^{TAG} mutants. We then performed western blots with anti-FLAG antibody which allowed us to see the amount of truncated vs. full length protein. These results are shown in Supplementary Figure 2 (c,d,f). In addition, full data set that was used for the analysis of both NFL-FLAG and FLAG-NFL Western blots is shown in Supplementary Table 1. As we also

discuss in the revised manuscript, it is important to note that the amount of truncated fragments will depend on the selected TAG position and can be reduced by improving the amber codon efficiency, for example by co-transfection with eRF1^{E55D}. This type of control experiment can be used to select the mutant with most optimal expression of the full-length protein, which would then be an additional parameter for selection of the best-performing TAG mutant. As already pointed out by the reviewer, we did not notice a negative effect of the NFL^{K363TAG} fragment on the formation of the neurofilament network, even though these new results show that it is present in the cell. However, this was not the case with the NFL^{K468TAG} mutant where the fragment itself or the K468TAG mutation might have affected formation of the network as we discuss in the manuscript. In addition to discussing these new data in the corresponding results section (lines 136-146), we discuss this limitation in the revised discussion (lines 492-499).

MY ORIGINAL COMMENT: There is no characterization or quantification of the efficiency of the CRISP/Cas9 modification, so it is unclear how efficient or specific it is.

MY RESPONSE TO THE REVISED MANUSCRIPT: One line 286 of the revised manuscript the authors write “Because of the limited efficiency of amber codon suppression, site-specific introduction of TAG at the position K363 would result in knock-down of the endogenous NFL. Therefore, we changed the approach by adding a hexapeptide linker and a 3xFLAG tag at the C terminus of NFL (Fig. 6a).” The use of the C-terminal tag ensures that the authors will only see full-length protein, but it does not solve the potential problem of partial knockdown of NFL in these cells. The authors have not shown what the effects are on the overall level of NFL in the cells. If the labeling is accompanied by partial knockdown, this would limit the general utility of the authors’ approach.

We have now edited the quoted sentence and the corresponding paragraph to avoid confusion (lines 298 – 303 in the revised manuscript). Introducing a stop codon at the position K363 would lead to NFL knock-down. To avoid this, instead of introducing TAG at the position K363, we introduced a TAG-containing linker and a 3xFLAG tag at the C terminus of NFL. Upon transcription, Nefl mRNA will contain full NFL coding sequence followed by a linker with UAG stop codon and 3xFLAG tag. During translation, following can happen: i) in the case of successful UAA incorporation, full-length NFL with the addition of linker-UAA-3xFLAG tag will be synthesized; ii) in the case of unsuccessful UAA incorporation or UAA absence, translation will stop at the UAG codon, located in the linker, downstream of the NFL coding sequence. In either case, the full-length NFL will be translated and the overall level of endogenous NFL will not be affected. This is also depicted in Figure 6c.

On line 292 of the revised manuscript, the authors write “The specificity was also confirmed by sequencing of the PCR product flanking the integration sites in genomic DNA isolated from transfected neurons. The efficiency of the 293 knock-in 6 days after transfection is around 20% (Extended Data Fig. 14c).” The sequencing data is not shown so the specificity is still unclear. However, this is a lesser concern for me as the specificity of CRISPR is not central to this study.

We have now added the sequencing results to the Extended Data Figures for both ORANGE (Extended Data Fig 14) and TKIT (Extended Data Fig 16) technologies.

MY ORIGINAL COMMENT: In Fig. 3 it is not clear that two different populations of NFL have been labeled because the two labels overlap. This likely reflects the turnover of subunit proteins within these polymers over the course of several days, but it is nonetheless not the best illustration of the potential of the approach. It would be more effective to apply the second label after just a few hours, instead of a few days, which might enable the authors to localize the site of NFL translation in the neuron, for example.

MY RESPONSE TO THE REVISED MANUSCRIPT: The authors have performed additional experiments and added new data to address this suggestion. I think it is a much better illustration of the utility of their approach.

We thank the reviewer for their comment.

MY ORIGINAL COMMENT: Fig. 5 seems to be of little utility. The authors state that it demonstrates nanoscale resolution but it is not clear what is being resolved. While the concept of dual-color pulse-chase experiments sounds appealing, it is not clear that it is revealing anything in this experiment.

MY RESPONSE TO THE REVISED MANUSCRIPT: The authors moved the figure to the supplement and increased the magnification in some of the panels. I am fine with this.

We thank the reviewer for their comment.

Reviewer #3 (Remarks to the Author):

The additional experiments and corrections made in the revised manuscript address my previous concerns. In my view, the manuscript has improved considerably and all issues were sufficiently clarified in the point-by-point response to reviewer comments. I do not have additional comments.

We thank the reviewer for their comments.

Reviewers' Comments:

Reviewer #1:

Remarks to the Author:

I appreciate the authors response to my comments and critiques and the considerable amount of thought and work that they have put into these revisions. I am satisfied with their response to all but the following two concerns. The second concern is one the authors can easily address.

However, the first concerns is a fundamental one in my opinion, and continues to undermine my confidence in the utility of their experimental strategy.

Concern #1: The low inefficiency of the amber codon suppression generates truncated proteins that could disrupt cellular function. This is a serious limitation of the authors' experimental strategy.

MY RESPONSE TO THE LATEST REVISIONS: The authors have added new data to show the proportion of truncated proteins generated by inefficient amber codon suppression. The data are of good quality and demonstrate that co-transfection with eRF1-E55D does improve the efficiency of codon suppression but even with this measure, approximately 45-68% of the NFL protein that is expressed is truncated. While I appreciate the quality of the authors data and the rigor and transparency with which they present it, I have to say that I still feel this is a significant weakness in their method. The approach is elegant, and this study represents a huge amount of work, but it is hard for me to see how this method can be adopted by others given this significant problem. The possibility of unknown (e.g., dominant negative) effects of the truncated proteins will hang like a cloud over any experiment that uses this methodology. For example, it is known from published work that truncated intermediate filament proteins can dominantly disrupt neurofilament structure and function in cells. These concerns would evaporate if someone figures out a way to make the amber codon suppression much more efficient, but until that point it is hard to see how the method adopted here will be useful.

Concern #2: The specificity of their genome editing is unclear.

MY RESPONSE TO THE LATEST REVISIONS: I appreciate the addition of the sequence information in Extended Data figs 14 and 16. However, it only raises more questions. It would have been helpful if the authors had marked the reading frame and the location of the UAG stop codons. It took this reviewer some time to understand, but in fig 14 it appears that there is a frame shift at the C-terminus due to a residue inserted during the genome editing, and also that the NFL sequence in fig 14 is truncated by about 6 amino acids. The sequencing traces in Fig 14 also reveal some complexity to the sequence in the region of the 3xFLAG tag which appears to imply a mixture of DNA sequences arising from the genome editing. I am concerned that none of this complexity is addressed in the text or figure legends. This is not a fundamental concern, as the specificity of CRISPR is not central to this study, but I do feel that the authors should annotate the sequence data better, and explicitly acknowledge the frame shift and truncation.

**Arsić et al_revision#3**

*Reviewer's comments are in black. Our response is in blue.*

**Reviewers' comments:**

Reviewer #1 (Remarks to the Author):

I appreciate the authors response to my comments and critiques and the considerable amount of
thought and work that they have put into these revisions. I am satisfied with their response to all but
the following two concerns. The second concern is one the authors can easily address. However, the
first concerns is a fundamental one in my opinion, and continues to undermine my confidence in the
utility of their experimental strategy.

*We thank the reviewer for their critical feedback and insightful comments throughout the whole review
process. However, as we disagree with the reasons for the rejection of our manuscript, we would like to
address these two concerns further. As we'll discuss below, there is no evidence that concern #1 is
undermining any of our findings. We also offer a solution to this problem. Concern #2 was discussed by
16 us already in the 1st version of the manuscript. Moreover, we overcame this concern by introducing
another genome engineering technology during the revision process.*

Concern #1: The low inefficiency of the amber codon suppression generates truncated proteins that
could disrupt cellular function. This is a serious limitation of the authors' experimental strategy.

*As we have already discussed, we agree that this is a limitation of the method, but we disagree with it
being the reason for the rejection of our manuscript. We have listed examples from the literature and
we have performed additional experiments to address this. Our new quantifications and results (lines
69-79 and 80-104 in this document; revised Extended Data Fig. 2d-f; new Extended Data Fig. 3 and
Supplementary Table 2 in the revised manuscript) confirm that the efficiency of the amber codon
suppression can be further increased. This results in significantly lower (almost undetectable) levels of
truncated protein.*

MY RESPONSE TO THE LATEST REVISIONS: The authors have added new data to show the proportion of
truncated proteins generated by inefficient amber codon suppression. The data are of good quality and
demonstrate that co-transfection with eRF1-E55D does improve the efficiency of codon suppression but
even with this measure, approximately 45-68% of the NFL protein that is expressed is truncated. While I
appreciate the quality of the authors data and the rigor and transparency with which they present it, I
have to say that I still feel this is a significant weakness in their method. The approach is elegant, and
this study represents a huge amount of work, but it is hard for me to see how this method can be
adopted by others given this significant problem. The possibility of unknown (e.g., dominant negative)
effects of the truncated proteins will hang like a cloud over any experiment that uses this methodology.
For example, it is known from published work that truncated intermediate filament proteins can
dominantly disrupt neurofilament structure and function in cells. These concerns would evaporate if

someone figures out a way to make the amber codon suppression much more efficient, but until that
point it is hard to see how the method adopted here will be useful.

We agree with the reviewer that the suboptimal efficiency of amber codon suppression is a limitation of
the method. We also need to add that this limitation applies to the labeling of any protein of interest
and not only neurofilament light chain. However, a plethora of published *in vitro* and *in vivo* studies that
have successfully used amber codon suppression speaks against this being a critical weakness of the
method. Furthermore, the combination of amber codon suppression and click chemistry has already
found additional application in neurobiology as demonstrated in a recent manuscript that was just
published in Nature Communications (<https://doi.org/10.1038/s41467-021-27025-w>). In this work,
which first appeared as a preprint during the revision of our work and which we cite in the revised
manuscript (lines 406-408), click chemistry was used to label transmembrane AMPAR regulatory
proteins (TARPs) further highlighting the potential of this method. A growing interest in this method is
another reason why we have discussed amber codon suppression efficiency limitation transparently
since the first version of the manuscript. This is also the reason why we did the click labeling of
endogenous NFL with a small C-terminal tag instead of the site-specific labeling. Every method has its
advantages and disadvantages; and whether the presence of truncated protein can be a problem or not,
depending on how the cell deals with it, is in-depth addressed here and in the revised manuscript.
Keeping in mind all the data (including the latest experiments) that we provide, we think that too much
weight is given to this concern and that it should not be the reason for the rejection of our manuscript.
Especially when keeping in mind positive comments of the other two reviewers.

In addition, the level of truncated and full-length proteins as measured by western blot in these types of
experiments is not absolute. Western blot itself is a semi-quantitative method. The amount of detected
proteins will depend on the expression host, transfection efficiency, western blot procedure (antibody
staining, exposure time, background) etc. and will also differ from experiment to experiment. In
addition, the amount of expressed proteins (fragment vs. full-length) will also differ from cell to cell and
western blot does not truly reflect what is happening at the level of individual cells. As our work relies
on transfections, X% of truncated protein as detected on the western blot does not necessarily mean
that each individual cell will express X% of truncated fragment and (100-X)% of the full-length protein.
Depending on the amount of the orthogonal PylRS and tRNA^{Pyl} that are necessary for the amber codon
suppression, individual cells might express more or fewer protein fragments. We have also noticed that
the way we presented our data in the previous version of the manuscript might be confusing. The values
showed amounts of fragments relative to the FLAG-NFL^{WT}. The reviewer wrote in their comment that
45-68% of the expressed protein is truncated, but we are not sure where do these numbers come from.
We have now obtained respective values by measuring relative amounts of the full-length and truncated
fragments in each of the individual lanes (conditions). For the previously analysed FLAG-NFL^{K363TAG} blots,
in the absence of eRF1^{E55D}, the amount of truncated protein was around 65%. In presence of eRF1^{E55D},
the amount was reduced to the following values (13%, 21%, 26%) in three different western blots. 13-
26% is significantly lower than lower than what was estimated by calculating the percentage of the
FLAG-NFL^{WT}. **We have included these quantifications in the revised manuscript (revised Extended Data
Figure 2f and new Supplementary Table 2).**

Furthermore, as we have already discussed in
 the manuscript, by using improved systems for
 amber codon suppression, its efficiency can be
 further increased. We already showed the
 effect of eRF1^{E55D} and we now performed new
 experiments with additional constructs for
 amber codon suppression. Based on the
 literature, we tested a plasmid that contains
 multiple copies of a designer (M15) tRNA
 (Serfling et al., 2018), as well as our codon
 optimized plasmid that we used for amber
 codon suppression in combination with
 genome engineering. Both of these plasmids
 show a significant reduction of the fragment
 amount as can be seen in the western blot on
 the right. In these conditions, fragments can
 barely be detected. Quantitative analysis
 shows that they correspond to less than 10% and 3% of the expressed NFL^{K363TAG}, respectively. This
 experimentally supports our claim that limited efficiency problem can be solved easily (in the previous
 version of the manuscript we wrote following: “In addition, the sequence-dependant varying levels of
 amber codon suppression efficiency result in the translation of protein fragments that could potentially
 affect cellular processes. However, by analyzing different TAG mutants and using more efficient
 approaches for amber codon suppression, the amount of truncated proteins can be reduced, as we and
 others have shown”). **We have included these data in our new Extended Data Figure 3. We have also
 updated corresponding results (lines 147-158) and discussion (lines 524-538) sections.**

Additional evidence in support of our claim that limited efficiency of amber codon suppression does not
 represent a fundamental concern when it comes to NFL labelling are following:

Our transfected neurons have normal morphology, neurons express our constructs over the course of
 several days without any signs of cytotoxicity, neurofilament network looks normal, and our click
 labelling co-localizes with the anti-NFL signal. If our method was affecting the function of NFL, we would
 have to see some problems. The only way for us to judge if neurofilaments are functioning properly is to
 look at the above-mentioned parameters. Other researchers in the field do the same. They transfect
 their constructs, such as fluorescent protein fusions, and they look at the cell morphology, overall health
 and the competence of their constructs to assemble into neurofilament network. This is nicely
 described in the manuscript that we cite (Uchida et al., 2016). Among other things, the authors also
 discuss problems with certain fluorescent protein fusions, such as mCherry. Although there is no
 explanation for these observations and we, for example, cannot understand why would mCherry and
 GFP fusions behave differently, the authors write the following: “mCherry has good brightness and
 photostability though in our hands mCherry-neurofilament fusions sometimes have a slight tendency to
 form small non-filamentous aggregates. If this problem is encountered, then TagRFP-T is a suitable
 alternative that does not exhibit this tendency”. In another manuscript from the same group

(https://www.nature.com/articles/ncb0300_137) from 2000, the following is written: “Fusion of GFP to
the carboxy terminus of NFM resulted in the expression of a fusion protein that formed non-filamentous
aggregates (data not shown); we believe that this may explain the punctate non-filamentous distribution
of C-terminally fused GFP–NFM observed by Yabe et al.²⁶ These data indicate that the presence of the
GFP domain at the N terminus of NFM does not interfere with its assembly properties, and that the N-
terminally fused fusion protein assembled into neurofilaments throughout the neuron.” In all the
published work since 2000, the only way to test NFL, NFM and NFH constructs was the appearance of
the neurofilament network and the presence/absence of aggregates. This is also how disease-related
mutants are tested and how mutants with dominant-negative effects were described (e.g.
<https://doi.org/10.1093/hmg/ddh236>; <https://doi.org/10.1002/cm.21566>;
<https://doi.org/10.1093/hmg/11.23.2837>; <https://doi.org/10.1242/jcs.00148>; <https://doi.org/10.1093/hmg/ddm272>).

Since we and others have no other way to test the function of NFL protein, then we should question any
other protein labelling methodology and its potential effect on the cellular function. In other words, if
we were the first to make NFL-fluorescent protein fusions, we would have a similar problem. We do not
expect truncated proteins in this case, but fluorescent protein itself can affect the function of NFL as
described in the literature and examples that we listed above. However, fluorescent protein fusions of
neurofilaments are accepted in the field and none of these physiological concerns stop the researchers
from performing their experiments and publishing their work. Since any labelling method could affect
cell health, we believe that different fluorescent labelling methods are complementary. And the more
methods we have, the researchers can find the most suitable one for their experiments. We also show
that not all of our tested constructs were physiologically relevant. For example, one of our labelling
mutants (R438TAG) clearly had an effect on the neurofilaments assembly and potentially cell health and
we discussed that already in the first version of the manuscript. But as we describe above, there is no
evidence that cell function is compromised in any way when using the NFL^{K363TAG} construct. The reviewer
#1 also wrote in one of their previous referee reports that we are clearly not seeing a disruption of
neurofilament assembly in our data. In addition, our new data offers a solution on how to reduce the
amount of truncated fragment to almost undetectable levels, diminishing the possibility of fragment-
induced disruptions. **We have updated our results and discussion sections to include these points**
**(lines 183-186 and 524-538).**

Concern #2: The specificity of their genome editing is unclear.

As we will discuss below, this concern has already been addressed but it seems that some of our
explanations got overlooked during the review process.

MY RESPONSE TO THE LATEST REVISIONS: I appreciate the addition of the sequence information in
Extended Data figs 14 and 16. However, it only raises more questions. It would have been helpful if the
authors had marked the reading frame and the location of the UAG stop codons. It took this reviewer
some time to understand, but in fig 14 it appears that there is a frame shift at the C-terminus due to a
residue inserted during the genome editing, and also that the NFL sequence in fig 14 is truncated by
about 6 amino acids. The sequencing traces in Fig 14 also reveal some complexity to the sequence in the

region of the 3xFLAG tag which appears to imply a mixture of DNA sequences arising from the genome
editing. I am concerned that none of this complexity is addressed in the text or figure legends. This is not
a fundamental concern, as the specificity of CRISPR is not central to this study, but I do feel that the
authors should annotate the sequence data better, and explicitly acknowledge the frame shift and
truncation.

We appreciate the dedication with which the reviewer approached this concern but we are surprised
that some of our explanations went unnoticed. Since the first version of the manuscript, we have
discussed the limitations of the ORANGE approach for the *Nefl* genome editing (data shown in Fig. 6 and
what is now Supplementary Fig. 15). The issue with the deletion of 6 amino acids (as a consequence of
limited number of PAM sites available for the genome editing of the *Nefl* gene) has been discussed
already in the first version of the manuscript (*lines 335-338 in the previous manuscript version where*
*following was written “Furthermore, it is important to note that when using the ORANGE knock-in*
*strategy for endogenous NFL tagging, the availability of protospacer adjacent motif (PAM) sites was a*
*limiting factor. Since the only suitable PAM site at the end of the Nefl gene is located upstream of the*
*stop codon, our approach resulted in deletion of six C-terminal amino acids”*) as well as the
corresponding methods sections). We also highlighted the one nucleotide insertion in the corresponding
Supplementary Figure and mentioned it in the figure legend, but we did not specify that this arises from
the mixture of genomic DNA sequences. The fact that we saw NFL-FLAG translated in the frame shows
that the insertion is not present in all of the neurons. But we agree that this could lead to confusion and
to avoid it, we updated the corresponding supplementary figures (Extended Data Fig. 15 and 17) to
include the open reading frame annotation. We also updated the corresponding results section (*lines*
*320 – 328; 358-359* in the revised manuscript). However, all of this is less relevant since we established
an alternative genome editing method (TKIT). TKIT method does not have these issues as we discussed
in the manuscript since it relies on the editing of noncoding regions (*as described in lines 338-240 of the*
*previous manuscript version and lines 358 - 362 of the current manuscript version where the following is*
*written: “This limitation can be avoided by using alternative genome engineering strategies that rely on*
*targeting of noncoding regions, and which offer more flexibility in choosing the target sites, such as*
*Targeted Knock-In with Two guides (TKIT) strategy”; lines 370 -374 where following is written “As the*
*editing happens in the intron and 3’ UTR, the main advantage of TKIT approach over ORANGE is the*
*preservation of the intact Nefl open reading frame”*). It is also important to mention that we established
TKIT approach during the first round of revision even without any of the reviewers' raising the concern
about 6 amino acid deletion or any other problem with ORANGE method. Our motivation was to
combine click labeling with the genome engineering approach which would allow us to keep the open
reading frame intact. The reason for this being that there is a lot of interest in the field in combining
these techniques (including the above-mentioned manuscript about click labeling of transmembrane
AMPAR regulatory proteins). Correspondingly, we have updated our discussion section to include
following text (*lines 448-457*): *“It is important to note that TKIT genome editing approach, as opposed to*
*ORANGE, allowed us to have an intact Nefl open reading frame (ORF). Since it does not rely on editing in*
*coding regions, TKIT could be used for site-specific incorporation of TAG codon in the ORF of the target*
*protein and subsequent site-specific click labelling of endogenous proteins. However, such premature*
*stop codons would almost certainly lead to the reduction of endogenous protein levels. This knock-down*

*would be caused partially by the suboptimal amber codon suppression efficiency and partially by the*
*nonsense-mediated decay of premature stop codon-containing mRNAs. For this reason, C-terminal*
*addition of TAG-containing small linkers and tags, as described here, represents a more optimal strategy*
*when combining amber codon suppression and genome editing.”*

Reviewers' Comments:

Reviewer #1:

Remarks to the Author:

Concern #1: The low inefficiency of the amber codon suppression generates truncated proteins that could disrupt cellular function. This is a serious limitation of the authors' experimental strategy.

MY RESPONSE TO THE LATEST REVISIONS: In the previous version of the manuscript the authors' analysis in Extended Data Figure 2 and Supplementary Data Table 2 showed that the proportion of full-length protein expressed in the presence of eRF1E55D (for the N-terminal FLAG tag) was reduced by 44-82% relative to the wild type protein for the four mutants examined. In the latest submission, they have revised their quantification in Extended Data Figure 2 and Supplementary Table 2 to represent the level of truncated protein as a percentage of the total full length and truncated protein in each sample rather than relative to the wild type sample. This reduced the estimated fraction of truncated protein for some mutants and increased it for others, ranging in a proportion of full-length protein ranging from 24-98%. Overall, the total amount of protein is lower with all the mutants compared to wild type. If this reflects instability of the truncated proteins, then the new method will tend to overestimate the efficiency of codon suppression. On the other hand, if the truncated proteins turn over more rapidly, then that would reduce concerns about their potential toxicity. Moreover, the authors have now added additional data in Extended Data Figure 3 which shows that the efficiency of amber codon suppression can be further improved to achieve as little as 3% truncated protein for the K363TAG mutant. Combined with the fact that there are no overt effects on neurofilament assembly in cells transfected with this and other mutants, I feel the authors have addressed my concern sufficiently.

Concern #2: The specificity of their genome editing is unclear.

MY RESPONSE TO THE LATEST REVISIONS: I thank the authors for their detailed response. The authors' additional annotation in Extended Data Figures 15 and 17 and additional explanation in the Results section is clear and addresses my concern.

**Arsić et al_revision#4**

*Reviewer's comments are in black. Our response is in blue.*

**Reviewers' comments:**

Reviewer #1 (Remarks to the Author):

Concern #1: The low inefficiency of the amber codon suppression generates truncated proteins that
could disrupt cellular function. This is a serious limitation of the authors' experimental strategy.

MY RESPONSE TO THE LATEST REVISIONS: In the previous version of the manuscript the authors'
analysis in Extended Data Figure 2 and Supplementary Data Table 2 showed that the proportion of full-
length protein expressed in the presence of eRF1E55D (for the N-terminal FLAG tag) was reduced by 44-
82% relative to the wild type protein for the four mutants examined. In the latest submission, they have
revised their quantification in Extended Data Figure 2 and Supplementary Table 2 to represent the level
of truncated protein as a percentage of the total full length and truncated protein in each sample rather
than relative to the wild type sample. This reduced the estimated fraction of truncated protein for some
mutants and increased it for others, ranging in a proportion of full-length protein ranging from 24-98%.
Overall, the total amount of protein is lower with all the mutants compared to wild type. If this reflects
instability of the truncated proteins, then the new method will tend to overestimate the efficiency of
codon suppression. On the other hand, if the truncated proteins turn over more rapidly, then that would
reduce concerns about their potential toxicity. Moreover, the authors have now added additional data
in Extended Data Figure 3 which shows that the efficiency of amber codon suppression can be further
improved to achieve as little as 3% truncated protein for the K363TAG mutant. Combined with the fact
that there are no overt effects on neurofilament assembly in cells transfected with this and other
mutants, I feel the authors have addressed my concern sufficiently.

Concern #2: The specificity of their genome editing is unclear.

MY RESPONSE TO THE LATEST REVISIONS: I thank the authors for their detailed response. The authors'
additional annotation in Extended Data Figures 15 and 17 and additional explanation in the Results
section is clear and addresses my concern.

*We thank the reviewer for their comments.*